# Colistin kills bacteria by targeting lipopolysaccharide in the cytoplasmic membrane

**Akshay Sabnis[1], Katheryn LH Hagart[1], Anna Klöckner[1,2,3,4], Michele Becce[2,3,4], Lindsay E Evans[1,5], R Christopher D Furniss[1], Despoina AI Mavridou[6], Ronan Murphy[7,8], Molly M Stevens[2,3,4], Jane C Davies[7,8], Gérald J Larrouy-Maumus[1], Thomas B Clarke[1], Andrew M Edwards[1]***

[1]MRC Centre for Molecular Bacteriology and Infection, Imperial College London, London, United Kingdom; [2]Department of Bioengineering, Imperial College London, London, United Kingdom; [3]Department of Materials, Imperial College London, London, United Kingdom; [4]Institute of Biomedical Engineering, Imperial College London, London, United Kingdom; [5]Department of Chemistry, Imperial College London, Molecular Sciences Research Hub, London, United Kingdom; [6]Department of Molecular Biosciences, University of Texas at Austin, Austin, United States; [7]National Heart and Lung Institute, Imperial College London, London, United Kingdom; [8]Department of Paediatric Respiratory Medicine, Royal Brompton Hospital, London, United Kingdom

*For correspondence:
a.edwards@imperial.ac.uk

Competing interests: The authors declare that no competing interests exist.

**Abstract** Colistin is an antibiotic of last resort, but has poor efficacy and resistance is a growing problem. Whilst it is well established that colistin disrupts the bacterial outer membrane (OM) by selectively targeting lipopolysaccharide (LPS), it was unclear how this led to bacterial killing. We discovered that MCR-1 mediated colistin resistance in *Escherichia coli* is due to modified LPS at the cytoplasmic rather than OM. In doing so, we also demonstrated that colistin exerts bactericidal activity by targeting LPS in the cytoplasmic membrane (CM). We then exploited this information to devise a new therapeutic approach. Using the LPS transport inhibitor murepavadin, we were able to cause LPS accumulation in the CM of *Pseudomonas aeruginosa*, which resulted in increased susceptibility to colistin in vitro and improved treatment efficacy in vivo. These findings reveal new insight into the mechanism by which colistin kills bacteria, providing the foundations for novel approaches to enhance therapeutic outcomes.

## Introduction

The emergence of multi-drug-resistant Gram-negative pathogens such as *Escherichia coli*, *Klebsiella pneumoniae* and *Pseudomonas aeruginosa* has led to the increased use of polymyxin antibiotics, which are often the only viable last-resort therapeutic option (*Velkov et al., 2010*; *Garg et al., 2017*; *Biswas et al., 2012*; *Thomas et al., 2019*). Two closely related polymyxin antibiotics are used clinically, colistin (polymyxin E) and polymyxin B, which share a high degree of structural similarity, consisting of a cationic peptide ring of 7 amino acids connected to a hydrophobic acyl tail by a linear chain of three amino acids (*Velkov et al., 2010*; *Biswas et al., 2012*).

Polymyxins are rapidly bactericidal towards Gram-negative bacteria in vitro but are considerably less efficacious in vivo, with up to 70% of patients failing to respond to colistin treatment (*Falagas et al., 2010*; *Paul et al., 2018*; *Linden et al., 2003*). Restrictions on dosage due to the nephrotoxicity of polymyxins mean that only 50% of people with normal renal function achieve a

**eLife digest** Antibiotics are life-saving medicines, but many bacteria now have the ability to resist their effects. For some infections, all frontline antibiotics are now ineffective. To treat infections caused by these highly resistant bacteria, clinicians must use so-called 'antibiotics of last resort'. These antibiotics include a drug called colistin, which is moderately effective, but often fails to eradicate the infection. One of the challenges to making colistin more effective is that its mechanism is poorly understood.

Bacteria have two layers of protection against the outside world: an outer cell membrane and an inner cell membrane. To kill them, colistin must punch holes in both. First, it disrupts the outer membrane by interacting with molecules called lipopolysaccharides. But how it disrupts the inner membrane was unclear. Bacteria have evolved several different mechanisms that make them resistant to the effects of colistin. Sabnis et al. reasoned that understanding how these mechanisms protected bacteria could reveal how the antibiotic works to damage the inner cell membrane.

Sabnis et al. examined the effects of colistin on *Escherichia coli* bacteria with and without resistance to the antibiotic. Exposing these bacteria to colistin revealed that the antibiotic damages both layers of the cell surface in the same way, targeting lipopolysaccharide in the inner membrane as well as the outer membrane.

Next, Sabnis et al. used this new information to make colistin work better. They found that the effects of colistin were magnified when it was combined with the experimental antibiotic murepavadin, which caused lipopolysaccharide to build up at the inner membrane. This allowed colistin to punch more holes through the inner membrane, making colistin more effective at killing bacteria. To find out whether this combination of colistin and murepavadin could work as a clinical treatment, Sabnis et al. tested it on mice with *Pseudomonas aeruginosa* infections in their lungs. Colistin was much better at killing *Pseudomonas aeruginosa* and treating infections when combined with murepavadin than it was on its own.

*Pseudomonas aeruginosa* bacteria can cause infections in the lungs of people with cystic fibrosis. At the moment, patients receive colistin in an inhaled form to treat these infections, but it is not always successful. The second drug used in this study, murepavadin, is about to enter clinical trials as an inhaled treatment for lung infections too. If the trial is successful, it may be possible to use both drugs in combination to treat lung infections in people with cystic fibrosis.

steady state serum concentration sufficient to kill bacteria (*Tran et al., 2016*; *Satlin et al., 2020*). As such, there is a desperate need to develop new approaches to enhance the efficacy of polymyxin antibiotics.

Barriers to increasing polymyxin efficacy include the significant gaps in our understanding of their mode of action. Whilst it is well established that the binding of polymyxins to lipopolysaccharide (LPS) on the surface of Gram-negative bacteria leads to disruption of the outer membrane (OM), it is unclear how this results in cell lysis and bacterial death (*Figure 1—figure supplement 1*; *Biswas et al., 2012*; *MacNair et al., 2018*). It is hypothesised that damage to the LPS monolayer enables polymyxins to traverse the OM via a process of 'self-directed uptake', although this has not been demonstrated experimentally (*MacNair et al., 2018*; *Powers and Hancock, 2003*). Once across the OM, polymyxins permeabilise the cytoplasmic membrane (CM), which is required for bacterial lysis and killing (*Velkov et al., 2010*; *Garg et al., 2017*; *Biswas et al., 2012*). However, the mechanism by which colistin damages the CM is unclear (*Powers and Hancock, 2003*; *Trimble et al., 2016*). It has been proposed that the surfactant activity of polymyxins, conferred by the positively charged peptide ring and hydrophobic tail, is sufficient to compromise the phospholipid bilayer of the CM via a detergent-like effect (*Velkov et al., 2010*; *Biswas et al., 2012*). In support of this, polymyxins can interact with mammalian cell membranes, leading to changes in epithelial monolayer permeability (*Berg et al., 1996*). Polymyxin antibiotics also have some inhibitory activity against the Gram-positive bacterium *Streptococcus pyogenes*, where the CM is formed of a phospholipid bilayer (*Betts et al., 2016*).

However, several lines of evidence call into doubt the ability of physiologically relevant concentrations of polymyxins to disrupt phospholipid bilayers. Firstly, the concentrations of polymyxin B

required to disrupt mammalian epithelial cells or inhibit the growth of *S. pyogenes* (8–16 μg ml⁻¹) are above typical serum concentrations of the antibiotic, and colistin at clinically relevant concentrations displays no activity against other Gram-positive organisms such as *Staphylococcus aureus* or *Enterococcus faecalis* (*Berg et al., 1996*; *Betts et al., 2016*; *Si et al., 2018*; *Kouidhi et al., 2011*). Furthermore, colistin has very little activity against synthetic phospholipid bilayer membranes unless LPS is present, a finding that explains why polymyxins are 30–100-fold less active against colistin-resistant *Acinetobacter baumanii* isolates that are LPS-deficient, with an OM composed of a phospholipid bilayer (*Khadka et al., 2018*; *Moffatt et al., 2010*; *Zhang et al., 2018*). Finally, molecular dynamics simulations show that the interaction of colistin with phospholipid bilayers is unlike what has been reported for other antimicrobial peptides that target phospholipid bilayers (*Fu et al., 2020*). Together, these observations call into question whether, at physiologically relevant concentrations, colistin disrupts the CM of Gram-negative bacteria via the engagement of the polymyxin antibiotic with membrane phospholipids.

In addition to the mode of action of colistin, there are also gaps in our understanding of the mechanisms by which colistin resistance protects bacteria from polymyxin antibiotics. In Gram-negative bacteria, LPS is synthesised in the cytoplasm via the Raetz pathway, during which it is introduced into the inner leaflet of the CM (*Raetz et al., 2007*; *Simpson and Trent, 2019*). It is then flipped to the outer leaflet of the CM by MsbA before transportation to the OM via the LptABCDEFG machinery (*Okuda et al., 2016*; *Zhou et al., 1998*; *Li et al., 2019*). To date, 10 mobile colistin resistance (*mcr*) gene variants have been described, all of which encode phosphoethanolamine (pEtN) transferases that modify the lipid A component of LPS with pEtN as it is trafficked through the CM on the way to the OM (*Liu et al., 2016*; *Carroll et al., 2019*; *Nang et al., 2019*; *Skov and Monnet, 2016*; *Wang et al., 2020*). Colistin resistance can also arise via mutations in genes encoding two-component regulatory systems such as PhoPQ, PmrAB, or BasRS (*Poirel et al., 2017*; *Janssen et al., 2020*). This typically leads to the addition of 4-amino-4-deoxy-l-arabinose (L-ara4N) and/or pEtN groups to LPS, with this modification also occurring at the CM (*Simpson and Trent, 2019*; *Poirel et al., 2017*).

Despite the association between MCR-mediated LPS modification and colistin resistance, there is evidence that it does not prevent polymyxin-mediated damage of the OM. For example, colistin has been shown to permit ingress of the *N*-phenyl-1-napthylamine (NPN) fluorophore into the OM of *E. coli* expressing *mcr*-1 (*MacNair et al., 2018*). Furthermore, colistin greatly enhances the activity of hydrophobic antibiotics such as rifampicin against polymyxin-resistant bacteria via disruption of the OM (*Brennan-Krohn et al., 2018*). However, despite colistin damaging the OM of resistant bacteria, it is unable to kill or lyse them (*MacNair et al., 2018*). This suggests that the modification of LPS with pEtN and/or L-ara4N protects the CM from colistin, but it is not clear how (*MacNair et al., 2018*; *Brennan-Krohn et al., 2018*).

Improving our knowledge of how colistin kills bacteria is essential to help devise new approaches to enhance the efficacy of last resort polymyxin antibiotics (*Liu et al., 2016*; *Carroll et al., 2019*; *Nang et al., 2019*; *Skov and Monnet, 2016*; *Wang et al., 2020*). To do this, we set out to better understand how *mcr*-1 protects bacteria from colistin and to then use this information to elucidate the mode of action of colistin, with the ultimate aim of exploiting this information to improve colistin efficacy.

## Results

### MCR-1 protects the CM but not the OM from colistin-mediated disruption

The first issue we wanted to resolve was whether MCR-1 protected the CM and/or OM of bacteria from colistin. To do this, we used an isogenic *E. coli* MC1000 strain pair, one of which expresses *mcr*-1 from the IPTG-inducible vector pDM1 (*mcr*-1) to ensure consistent expression under our experimental conditions, and the other transformed with the pDM1 vector alone as a control (pEmpty) (*Dortet et al., 2018*; Key resources table). As expected, we found that *E. coli* MC1000 expressing *mcr-1* had a significantly greater colistin minimum inhibitory concentration (MIC, 2 μg ml⁻¹) compared to the MC1000 pEmpty control strain (0.25 μg ml⁻¹), which was similar to that seen

for clinical isolates (*Dortet et al., 2018*; *Figure 1—figure supplement 2*). This confirmed that the *E. coli* cells were producing functional MCR-1.

To fully characterise the LPS-modifying activity of MCR-1, we undertook MALDI-TOF-based lipidomic analysis of both whole *E. coli* cells and *E. coli* spheroplasts that lacked an OM (*Weiss and Fraser, 1973*). We confirmed spheroplast formation by microscopy and used FITC labelling of OM surface proteins to demonstrate removal of the OM (*Figure 1—figure supplement 3*, *Figure 1—figure supplement 4*). Our lipidomic analysis revealed the presence of LPS modified with pEtN in both the CM and OM of *mcr*-1 expressing bacteria, consistent with the location of MCR-1 in the CM (*Dortet et al., 2018*; *Furniss et al., 2019*; *Figure 1—figure supplement 5*). Of note, whilst 42 ± 19% of total cellular LPS from MCR-1-producing *E. coli* was unmodified, the proportion of unmodified LPS in the CM was just 21 ± 2% (*Figure 1A*, *Figure 1—figure supplement 5*).

We next assessed the effect of colistin on the integrity of the *E. coli* OM using the hydrophobic NPN dye, which fluoresces upon contact with phospholipids exposed by damage to the LPS monolayer (*MacNair et al., 2018*; *Helander and Mattila-Sandholm, 2000*). As expected, colistin caused permeabilisation of the OM of the *E. coli* pEmpty strain in a dose-dependent manner (*Figure 1B*). In agreement with previous findings, we found that colistin also disrupted the OM of *E. coli* expressing *mcr*-1 to a similar degree to *E. coli* pEmpty (*Figure 1B*; *MacNair et al., 2018*). To further investigate permeabilisation of the OM by colistin, we assessed the susceptibility of bacteria to rifampicin in the presence of the polymyxin antibiotic. Rifampicin cannot normally cross the OM, which makes *E. coli* intrinsically resistant to the antibiotic. However, in keeping with previous work, we found that colistin sensitised *E. coli* expressing *mcr*-1 to rifampicin, with a fractional inhibitory concentration index (FICI) value of 0.14 indicating synergy between the two antibiotics in a checkerboard assay (*Figure 1—figure supplement 6*; *MacNair et al., 2018*). This confirmed that colistin disrupted the OM of resistant bacteria producing MCR-1. Therefore, MCR-1-mediated changes to LPS did not prevent permeabilisation of the OM by colistin, which reflects the presence of the relatively large quantity of unmodified LPS in the OM as determined in our lipidomic analysis (*Figure 1A*, *Figure 1—figure supplement 5*).

Next, we assessed damage to the CM structure in the *E. coli* strain pair during colistin exposure, using the membrane impermeant dye propidium iodide (PI). PI fluoresces upon contact with DNA in the bacterial cytoplasm, and thus is indicative of permeabilisation of the both the OM and CM of whole bacterial cells (*Allison and Lambert, 2015*; *Pietschmann et al., 2009*). As expected, colistin exposure resulted in a strong PI signal from *E. coli* pEmpty cells, indicative of CM permeabilisation (*Figure 1C*), which gradually declined, most likely due to nucleases released from lysed bacteria (*Lee et al., 2017*). However, despite colistin permeabilising the OM of *E. coli* expressing *mcr*-1, the CM of these bacteria remained intact, as demonstrated by the lack of PI-mediated fluorescence (*Figure 1C*). In keeping with these findings, colistin caused lysis of *E. coli* pEmpty cells, as seen by a reduction in $OD_{595nm}$ readings over time (*Figure 1D*). By contrast, *E. coli* cells producing MCR-1 grew in the presence of colistin despite the damage the polymyxin caused to the OM, as demonstrated by an increase in $OD_{595nm}$ measurements over time (*Figure 1D*). As such, the damage caused to the OM of *E. coli* MCR-1 strain by colistin is likely to be minor.

Taken together, these data demonstrate that MCR-1 protects the CM but not the OM from colistin-mediated permeabilisation.

## Colistin targets LPS in the CM

Although colistin was able to permeabilise the OM of *E. coli* expressing *mcr*-1, it was possible that the pEtN modifications might reduce the ability of the antibiotic to access the periplasm and thus the CM. To negate this possibility and focus on whether MCR-1 mediated LPS modification directly protected the CM from colistin, we performed experiments using spheroplasts of our *E. coli* strains that lacked both OM and cell wall.

To test whether LPS modifications altered the biophysical properties of the CM, we measured both membrane fluidity and surface charge of the *E. coli* spheroplasts using established methods. There were no differences in fluidity of the CM between *E. coli* pEmpty or *mcr-1*-expressing cells (*Figure 2—figure supplement 1*). As might be expected, there was a slight increase in the positive charge of the CM of the *mcr-1*-expressing *E. coli* relative to the pEmpty control, indicative of the presence of cationic pEtN modifications to LPS (*Figure 2—figure supplement 1*). To investigate whether this slight increase in membrane positivity was likely to be sufficient to repel colistin from

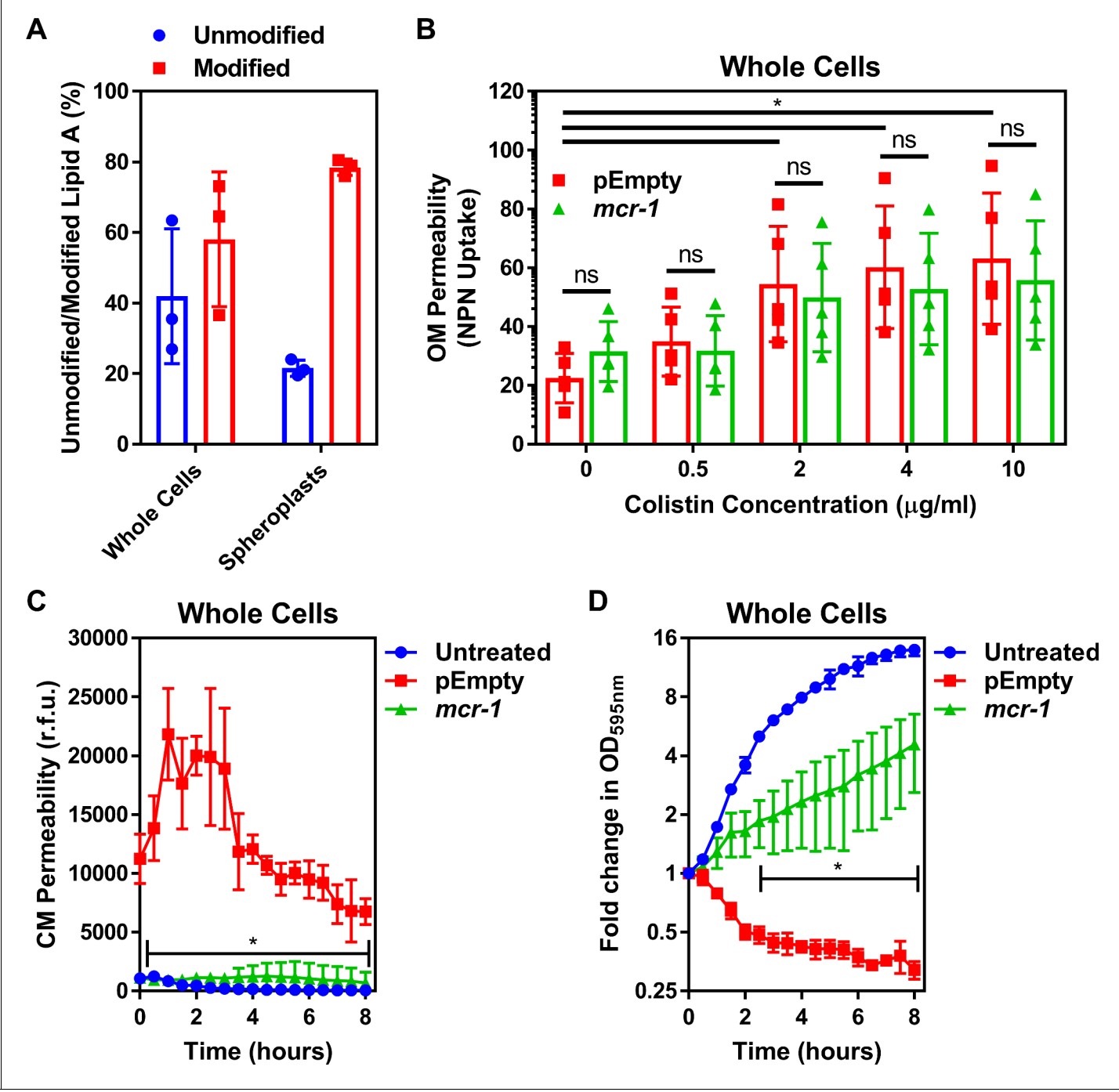

**Figure 1.** Colistin disrupts the outer membrane but not the cytoplasmic membrane of *E. coli* expressing *mcr-1*. (A) Quantification of LPS modified with phosphoethanolamine, expressed as the percentage of unmodified lipid A and unmodified lipid A, in whole cells and spheroplasts of *E. coli* MC1000 expressing *mcr-1*, as determined by MALDI-TOF-based lipidomics (n = 3 in duplicate, *p<0.05 between Whole Cells and Spheroplasts). (B) OM disruption of *E. coli* MC1000 cells expressing *mcr-1* or an empty plasmid control strain (pEmpty) during 10 min of exposure to colistin at the indicated antibiotic concentrations, as determined by uptake of the fluorescent dye NPN (10 μM) (n = 5, each data point represents the arithmetic mean of 20 replicate measurements; ns: p>0.05 between pEmpty and *mcr-1* strains, *p<0.05 between the indicated concentrations of colistin). (C) Permeabilisation of the CM of *E. coli* MC1000 cells expressing *mcr-1* or empty plasmid-containing cells during incubation with colistin (4 μg ml$^{-1}$), as determined using 2.5 μM propidium iodide (PI) (n = 4; *p<0.0001 between pEmpty and *mcr-1* strains). (D) Growth or lysis of *E. coli* MC1000 cells expressing *mcr-1* or empty plasmid control cells during exposure to colistin (4 μg ml$^{-1}$), as measured using OD$_{595nm}$ readings (n = 4; *p<0.05 between pEmpty and *mcr-1* strains). Data in (A) were analysed by a two-tailed paired Student's *t*-test. Data in (B–D) were analysed by a two-way ANOVA with Sidak's (B) or

*Figure 1 continued on next page*

*Figure 1 continued*

Dunnett's (**C, D**) post hoc tests. Data are presented as the arithmetic mean, and error bars represent the standard deviation of the mean. OM: outer membrane; NPN: *N*-phenyl-1-naphthylamine; CM: cytoplasmic membrane; r.f.u.: relative fluorescence units; OD: optical density.

The online version of this article includes the following figure supplement(s) for figure 1:

**Figure supplement 1.** Colistin causes outer membrane (OM) disruption, but the process by which this leads to cytoplasmic membrane (CM) damage and bacterial lysis is not known.

**Figure supplement 2.** Characterisation of the *E. coli* MC1000 strain harbouring a plasmid encoding the colistin resistance gene *mcr-1*, and an MC1000 strain containing the pDM1 plasmid only (pEmpty) as a control strain.

**Figure supplement 3.** Formation of *E. coli* pEmpty and *mcr-1* spheroplasts.

**Figure supplement 4.** Conversion of *E. coli* whole cells to spheroplasts results in removal of the OM, and no OM contamination in the CM.

**Figure supplement 5.** The ratio of modified lipid A to unmodified lipid A is significantly greater in the cytoplasmic membrane (CM) than in the outer membrane (OM) of *E. coli* expressing *mcr-1*.

**Figure supplement 6.** Colistin potentiates the activity of rifampicin against colistin-resistant *E. coli* expressing *mcr-1*.

the membrane, we determined the susceptibility of spheroplasts from *E. coli* MC1000 *mcr*-1 or pEmpty to colistin and compared it with the cationic antimicrobial peptides (CAMPs) daptomycin or nisin. Both CAMPs are well characterised for their ability to permeabilise phospholipid bilayers and, like colistin, they are positively charged, enabling us to detect whether the change to membrane charge conferred by MCR-1-modified LPS in the CM contributed specifically to polymyxin resistance (*Karas et al., 2020*; *Zendo et al., 2010*). Importantly, increased membrane positive charge is a common mechanism of resistance to daptomycin (*Karas et al., 2020*).

In the absence of treatment, there was a small but progressive loss of CM integrity over time due to the fragile nature of spheroplasts. However, allowing for this, the CM of spheroplasts of *E. coli* MC1000 *mcr*-1 was resistant to damage by colistin, but susceptible to daptomycin and nisin (*Figure 2A–C*). By contrast, colistin, daptomycin, and nisin all permeabilised the CM of spheroplasts of *E. coli* pEmpty (*Figure 2A–C*). In keeping with the data from assays measuring CM damage, colistin, daptomycin, and nisin all caused lysis of spheroplasts of *E. coli* pEmpty, whilst the spheroplasts from *E. coli* expressing *mcr-1* were undamaged by colistin, but were lysed by both daptomycin and nisin (*Figure 2D–F*). Combined, these data demonstrated that the protection afforded to the CM by MCR-1 is specific for colistin, and that the polymyxin antibiotic does not share the same target as the phospholipid-targeting CAMPs.

To further explore the specificity of MCR-1-mediated LPS modifications in the CM for protection against colistin, we produced spheroplasts of *E. coli* with different levels of LPS modification. This revealed a clear dose-dependent relationship between the abundance of unmodified LPS in the CM and the susceptibility of spheroplasts to colistin-mediated CM damage and lysis (*Figure 2—figure supplement 2*).

Therefore, since MCR-1 specifically modifies LPS and this modification selectively protects the CM from colistin in a dose-dependent manner, we concluded that LPS is the CM target of colistin, just as it is in the OM.

## Colistin damages the CM by disrupting cation bridges between LPS molecules

To understand how colistin targeting of LPS in the CM leads to membrane disruption, we studied the role of cation bridges which are crucial for stabilising interactions between LPS molecules, by exposing spheroplasts from *E. coli* pEmpty cells to colistin in the absence or presence of excess magnesium. In keeping with a role for cation bridges, we found that magnesium chloride conferred dose-dependent protection from colistin-mediated CM disruption (*Figure 3A*). To rule out a general protective osmotic effect from the higher salt concentration, we demonstrated that identical concentrations of sodium chloride did not protect spheroplasts from colistin (*Figure 3B*). Furthermore, the presence of exogenous magnesium had no significant effect on reducing spheroplast CM damage caused by daptomycin or nisin (*Figure 3C,D*), confirming that these CAMPs do not have the same CM target as colistin.

In conclusion, our findings demonstrate that colistin targets LPS in the CM of polymyxin-susceptible *E. coli*, leading to the displacement of cationic inter-LPS bridges, membrane disruption, and ultimately bacterial lysis. This is similar to the mechanism by which colistin disrupts the OM of bacteria

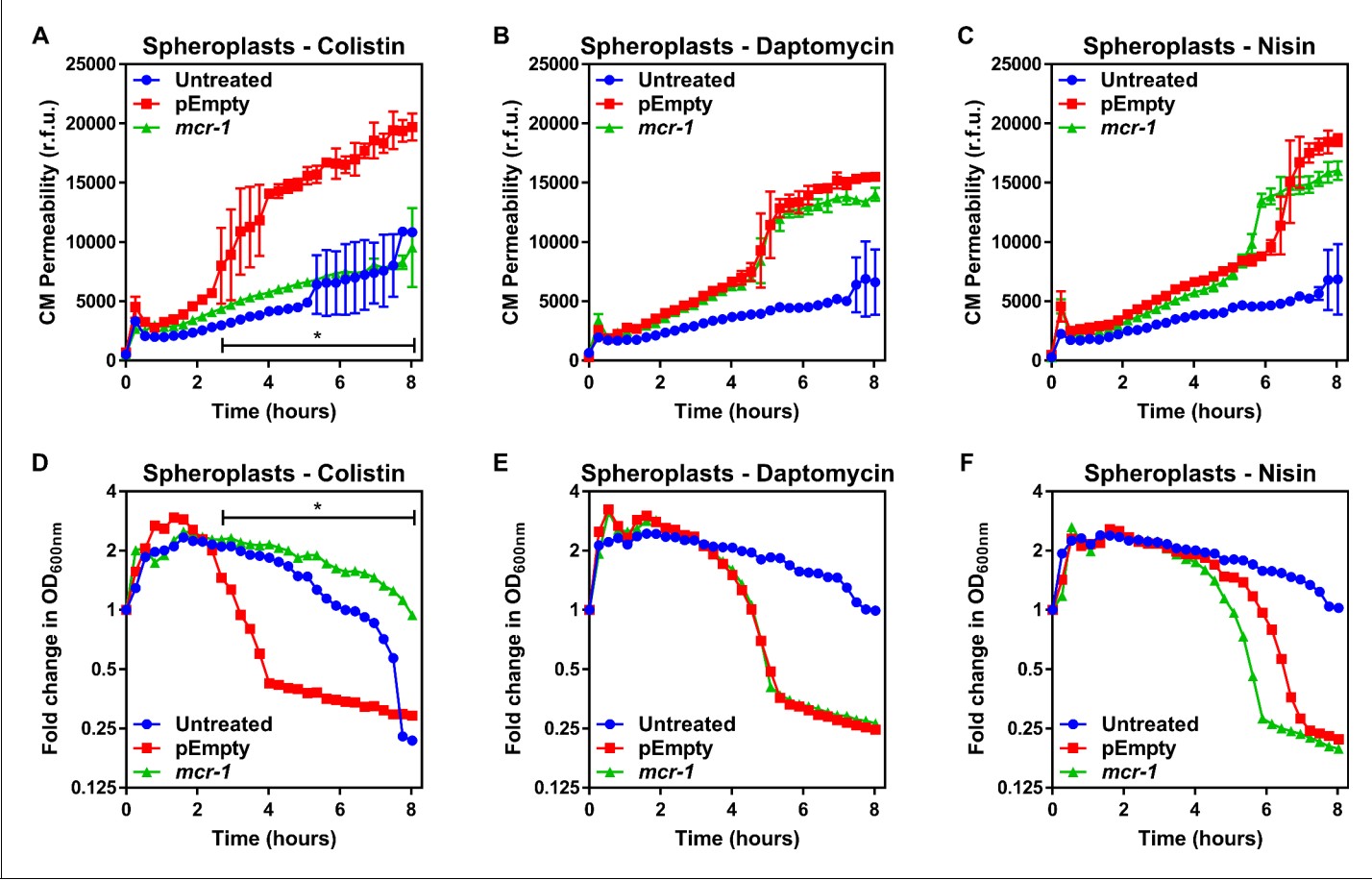

**Figure 2.** MCR-1 protects the cytoplasmic membrane of *E. coli* spheroplasts from colistin but not other cationic antimicrobial peptides. (A–C) Permeabilisation of the CM of *E. coli* MC1000 spheroplasts generated from bacteria expressing *mcr-1* or empty plasmid control bacteria (pEmpty) during incubation with (A) colistin (4 μg ml$^{-1}$), (B) daptomycin (20 μg ml$^{-1}$, with 1.25 mM Ca$^{2+}$ ions), or (C) nisin (20 μg ml$^{-1}$), as determined using 0.25 μM PI (n = 3, experiment performed on four independent occasions; *p<0.01 between pEmpty and *mcr-1* strains). (D–F) Lysis of *E. coli* MC1000 spheroplasts generated from bacteria expressing *mcr-1* or empty plasmid control bacteria during incubation with (D) colistin (4 μg ml$^{-1}$), (E) daptomycin (20 μg ml$^{-1}$, with 1.25 μM Ca$^{2+}$ ions), or (F) nisin (20 μg ml$^{-1}$), as measured using OD$_{600nm}$ readings (n = 3, experiment performed on four independent occasions; *p<0.05 between pEmpty and *mcr-1* strains, error bars are omitted for clarity). Data in (A–F) were analysed by a two-way ANOVA with Dunnett's post hoc test. Data are presented as the arithmetic mean, and error bars, where shown, represent the standard deviation of the mean. CM: cytoplasmic membrane; r.f.u.: relative fluorescence units; OD: optical density.

The online version of this article includes the following figure supplement(s) for figure 2:

**Figure supplement 1.** LPS modifications in the CM of colistin-resistant *E. coli* expressing *mcr-1* has a small effect on membrane charge but not membrane fluidity.

**Figure supplement 2.** The amount of unmodified LPS in the CM of colistin-resistant *E. coli* expressing *mcr-1* is proportional to the degree of susceptibility to colistin-mediated CM damage.

and synthetic phospholipid bilayer membranes containing low levels of LPS (*Khadka et al., 2018*; *Moore and Hancock, 1986*; *D'amato et al., 1975*). However, the high levels of LPS modified by pEtN in the CM of MCR-1-producing *E. coli* prevent colistin from targeting LPS in the CM, protecting the membrane and conferring resistance to the polymyxin antibiotic.

## Murepavadin-triggered LPS accumulation in the CM sensitises *P. aeruginosa* to colistin

Having determined that colistin kills bacteria by targeting LPS in the CM, we wanted to use this information to develop a new therapeutic approach to enhance colistin efficacy.

Murepavadin is a first-in-class peptide-based inhibitor of the LptD component of the LptABC-DEFG complex of *P. aeruginosa* that transports LPS from the CM to the OM (*Andolina et al., 2018*).

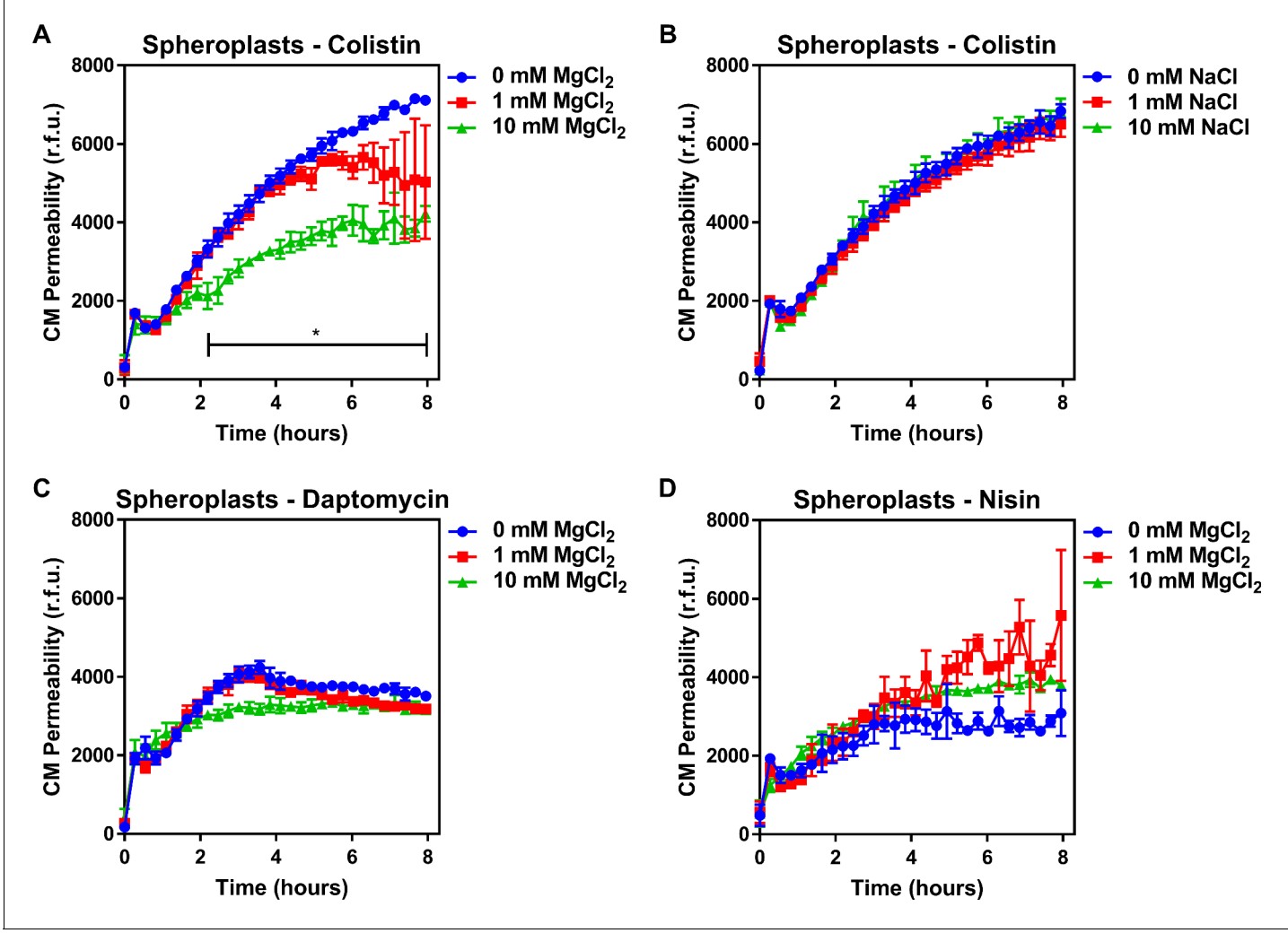

**Figure 3.** Colistin damages the cytoplasmic membrane by disrupting cation bridges between LPS molecules. (A, B) Permeabilisation of the CM of *E. coli* MC1000 spheroplasts generated from empty plasmid control bacteria during incubation with colistin (4 μg ml$^{-1}$), in the absence or presence of either MgCl$_2$( A) or NaCl (B) at the indicated concentrations, as determined using 0.25 μM PI (n = 3, experiment performed on three independent occasions; *p<0.01 between 0 mM MgCl$_2$ and 10 mM MgCl$_2$). (C, D) Permeabilisation of the CM of *E. coli* MC1000 spheroplasts generated from empty plasmid control bacteria during incubation with either (C) daptomycin (20 μg ml$^{-1}$, with 1.25 mM Ca$^{2+}$ ions) or (D) nisin (20 μg ml$^{-1}$), in the absence or presence of MgCl$_2$ at the indicated concentrations, as determined using 0.25 μM PI (n = 3, experiment performed on three independent occasions). Data in (A–D) were analysed by a two-way ANOVA with Dunnett's post hoc test. Data are presented as the arithmetic mean, and error bars represent the standard deviation of the mean. CM: cytoplasmic membrane; r.f.u.: relative fluorescence units.

Thus, inhibition of the Lpt system in *P. aeruginosa* leads to LPS accumulation in the CM, which we hypothesised would increase the susceptibility of the bacterium to colistin (*Andolina et al., 2018*; *Sperandeo et al., 2008*).

To test our hypothesis, we first used a checkerboard MIC assay and found that colistin synergised with murepavadin against *P. aeruginosa* PA14 cells (FICI value of 0.375), revealing that sub-lethal concentrations of the LptD inhibitor sensitised the bacterium to colistin (*Figure 4A*; *Odds, 2003*).

To confirm that sub-lethal concentrations of murepavadin altered LPS abundance in the CM, *P. aeruginosa* was incubated with murepavadin, before the amount of LPS in whole cells and spheroplasts was measured using the well-established Limulus Amoebocyte Lysate (LAL) assay, since previous work has shown this approach to be a highly accurate way of quantifying LPS in whole cell lysates (*Figure 4—figure supplement 1*, *Figure 4—figure supplement 2*; *Hoppe Parr et al., 2017*). The suitability of the LAL assay was further confirmed using a MALDI-TOF-based lipidomic analysis of spheroplast lysates, which confirmed that the LPS in both murepavadin-exposed and untreated

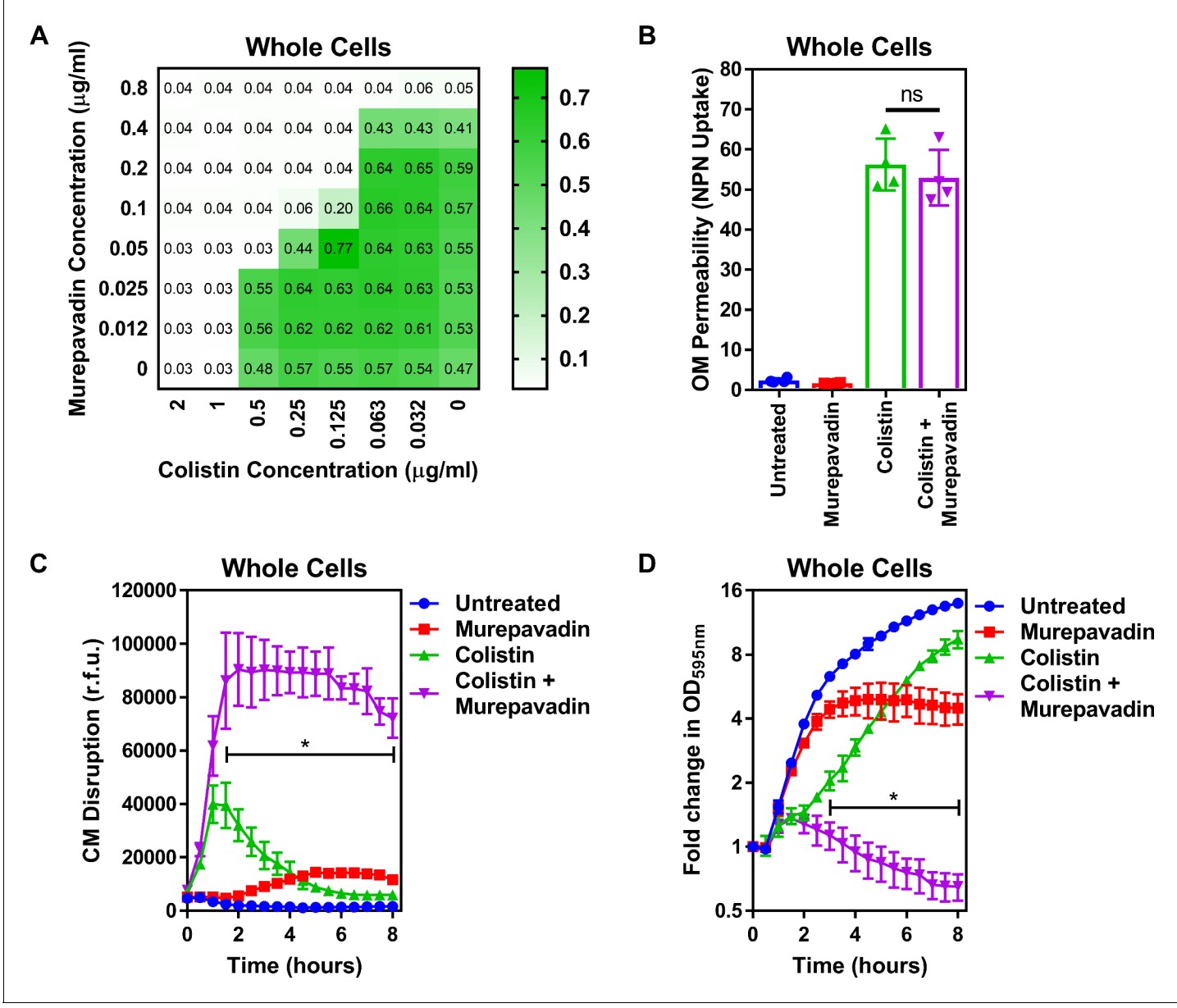

**Figure 4.** Murepavadin sensitises *P. aeruginosa* to colistin by increasing LPS abundance in the cytoplasmic membrane. (A) Checkerboard broth microdilution assay showing the synergistic growth-inhibitory interaction between colistin and the LPS transport inhibitor murepavadin against *P. aeruginosa* PA14 cells, as determined by measuring $OD_{595nm}$ after 18 hr incubation. (B) OM disruption of *P. aeruginosa* PA14 cells during 10 min exposure to colistin (2 µg ml$^{-1}$) in the absence or presence of murepavadin (0.05 µg ml$^{-1}$), as assessed by uptake of the fluorescent dye NPN (10 µM) (n = 4, each data point represents the arithmetic mean of 20 replicate measurements; ns: p>0.05 between colistin-treated bacteria). (C) CM disruption of *P. aeruginosa* PA14 cells exposed to colistin (2 µg ml$^{-1}$) in the absence or presence of murepavadin (0.05 µg ml$^{-1}$), as determined using 2.5 µM PI (n = 4; *p<0.0001 for colistin and murepavadin-exposed cells compared to colistin alone). (D) Lysis of *P. aeruginosa* PA14 cells exposed to colistin (2 µg ml$^{-1}$) in the absence of presence of murepavadin (0.05 µg ml$^{-1}$), as measured by $OD_{595nm}$ readings (n = 4; *p<0.01 for colistin and murepavadin-exposed cells compared to colistin alone). Data in (B) were analysed by a one-way ANOVA with Tukey's post hoc test. Data in (C, D) were analysed by a two-way ANOVA with Dunnett's post hoc test. Data are presented as the arithmetic mean, and error bars represent the standard deviation of the mean. OM: outer membrane; NPN: *N*-phenyl-1-naphthylamine; CM: cytoplasmic membrane; r.f.u.: relative fluorescence units; OD: optical density.

The online version of this article includes the following figure supplement(s) for figure 4:

**Figure supplement 1.** Formation of *P. aeruginosa* spheroplasts.

**Figure supplement 2.** Conversion of *P. aeruginosa* whole cells to spheroplasts results in removal of the OM, and no OM contamination in the CM.

**Figure supplement 3.** Murepavadin increases the abundance of LPS in the cytoplasmic membrane (CM) of *P. aeruginosa*.

**Figure supplement 4.** The LPS transport inhibitor murepavadin has no effect on reducing growth of *P. aeruginosa* at the concentration used.

**Figure supplement 5.** Polymyxin B nonapeptide (PMBN) does not display synergy with murepavadin against *P. aeruginosa* PA14.

*Figure 4 continued on next page*

*Figure 4 continued*

**Figure supplement 6.** PMBN–murepavadin combination therapy does not promote killing of *P. aeruginosa*.

**Figure supplement 7.** Murepavadin enhances the ability of colistin to damage the CM and trigger bacterial lysis.

bacteria was unmodified and thus able to be accurately detected and quantified (*Figure 4—figure supplement 3*; *Takayama et al., 1984*). Sub-lethal concentrations of murepavadin caused a slight reduction in LPS levels in the OM of *P. aeruginosa* cells, but a significant increase in the amount of LPS in the CM compared to untreated spheroplasts (*Figure 4—figure supplement 3*, *Figure 4—figure supplement 4*). Moreover, our lipidomic analysis revealed that the ratio of lipid A:phospholipid increased in *P. aeruginosa* spheroplasts pre-exposed to murepavadin, confirming that the LptD inhibitor caused LPS to accumulate in the CM (*Figure 4—figure supplement 3*).

Next, we proceeded to test whether LPS accumulation in the CM increased the susceptibility of *P. aeruginosa* to colistin. We started by examining the effect of colistin on the OM and CM of *P. aeruginosa* exposed, or not, to murepavadin. Despite the slight reduction of LPS at the OM caused by murepavadin, colistin permeabilised the OM to the same extent as bacteria that had not been exposed to murepavadin, with similar levels of NPN uptake (*Figure 4B*). By contrast, however, murepavadin significantly enhanced permeabilisation of the CM by colistin in whole cells of *P. aeruginosa*, as determined via uptake of PI (*Figure 4C*). Thus, an increase in LPS levels in the CM promoted colistin-mediated damage, in keeping with our conclusion that LPS in the CM is the target of the polymyxin antibiotic. Furthermore, *P. aeruginosa* cells exposed to murepavadin were more rapidly lysed by colistin than untreated cells (*Figure 4D*).

Taken together, these findings indicated that LPS accumulation in the CM increased susceptibility to the polymyxin antibiotic. However, an alternative explanation for the synergy between colistin and murepavadin was that polymyxin-mediated damage to the OM enabled murepavadin greater access to LptD in the periplasm. To test this, we first examined whether the OM permeabilising agent polymyxin B nonapeptide (PMBN) also synergised with murepavadin. However, we did not see synergy in MIC checkerboard or bactericidal assays between PMBN and murepavadin (*Figure 4—figure supplement 5*, *Figure 4—figure supplement 6*). Next, we pre-treated *P. aeruginosa* with murepavadin alone to cause LPS accumulation in the CM and then removed the murepavadin by washing before converting the whole cells to spheroplasts and exposing these to colistin alone. The murepavadin pre-treated spheroplasts were much more susceptible to colistin-induced CM damage and lysis than untreated spheroplasts (*Figure 4—figure supplement 7*). Therefore, colistin does not sensitise *P. aeruginosa* to murepavadin by compromising the OM; rather, murepavadin-induced accumulation of LPS in the CM potentiates the activity of colistin.

Taken together, these experiments provided further evidence that colistin targets LPS in the CM and suggest that murepavadin and colistin might form a useful combination therapy.

## Combination therapy with colistin and murepavadin promotes clearance of *P. aeruginosa*

*P. aeruginosa* is a major cause of chronic lung infection in people with cystic fibrosis (CF) and bronchiectasis. In both conditions, disease severity can rapidly increase during episodes of 'exacerbation' which must be treated aggressively to quickly restore lung function and reduce long-term damage (*Polverino et al., 2017*; *Karampitsakos et al., 2020*; *Stanford et al., 2021*). Therefore, having shown that murepavadin sensitised the CM to colistin-mediated damage, we wanted to determine whether this translated into enhanced antibacterial activity against relevant clinical isolates and increased treatment efficacy in vivo. We found that a sub-lethal concentration of murepavadin sensitised *P. aeruginosa* PA14 to a normally sub-lethal concentration of colistin (2 μg ml$^{-1}$), resulting in >10,000-fold reduction in c.f.u. counts relative to bacteria incubated with murepavadin or colistin alone after 8 hr (*Figure 5A*). We also found that murepavadin potentiated the activity of even lower concentrations of colistin (1 μg ml$^{-1}$), with exposure to the LPS transport inhibitor increasing the ability of the polymyxin antibiotic to damage the CM, triggering bacterial lysis and cell death (*Figure 5—figure supplement 1*).

We next examined a panel of 15 multi-drug resistant *P. aeruginosa* clinical strains, isolated from the sputum of CF patients, to investigate whether murepavadin increased colistin-mediated bacterial

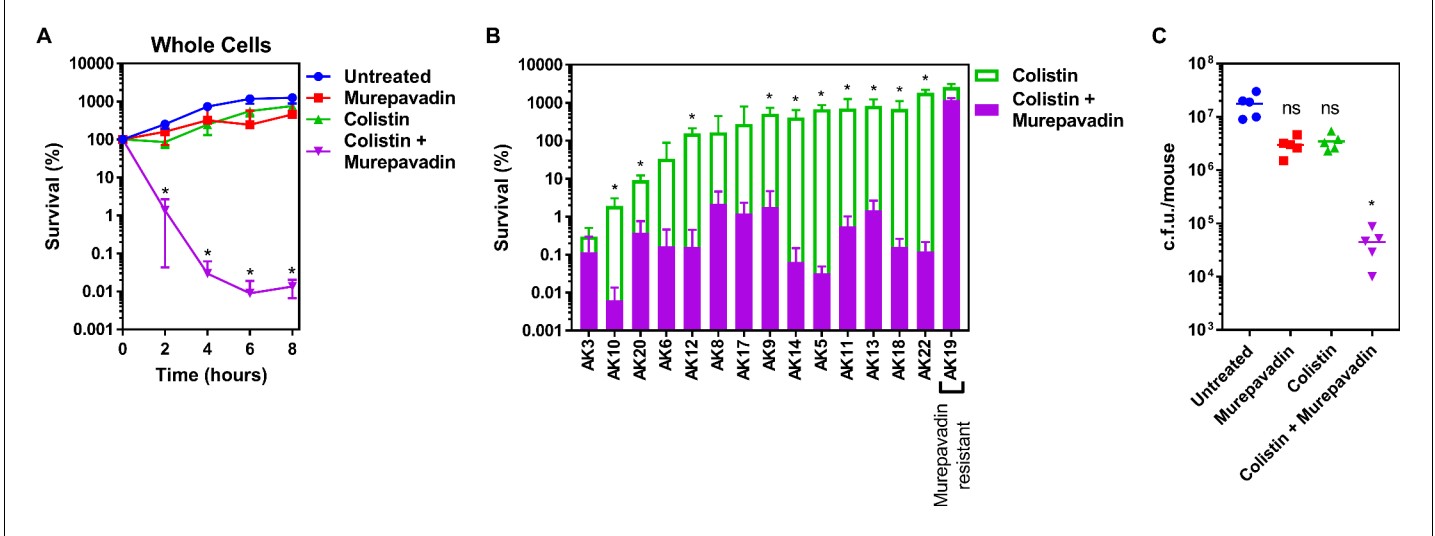

**Figure 5.** Colistin-murepavadin combination therapy promotes killing of *P. aeruginosa* in vitro and in vivo. (A) Survival of *P. aeruginosa* PA14 cells exposed to colistin (2 µg ml$^{-1}$) in the absence of presence of murepavadin (0.05 µg ml$^{-1}$), as determined by c.f.u. counts (n = 4; *p<0.05 for colistin and murepavadin-exposed cells compared to colistin alone). (B) Survival or growth of a panel of clinical multidrug-resistant *P. aeruginosa* strains isolated from the sputum of cystic fibrosis patients after 8 hr exposure to colistin (2 µg ml$^{-1}$) alone, or in the presence of a sub-lethal concentration (0.5× MIC) of murepavadin, as determined by c.f.u. counts (n = 4; *p<0.01 for colistin and murepavadin-exposed cells compared to colistin alone). (C) Burden of *P. aeruginosa* PA14 in the lungs of mice after 3 hr treatment with murepavadin (0.25 mg kg$^{-1}$), colistin (5 mg kg$^{-1}$), neither antibiotic, or both antibiotics in combination, as determined by c.f.u. counts (each data point represents a single mouse; for each group, n = 5; ns: p>0.05, *p<0.001 compared to untreated mice). Data in (A, B) were analysed by a two-way ANOVA with Dunnett's (A) or Sidak's (B) post hoc tests. Data in (C) were analysed by a Kruskal–Wallis test with Dunn's post hoc test. Data are presented as the arithmetic mean, and error bars, where shown, represent the standard deviation of the mean.

The online version of this article includes the following figure supplement(s) for figure 5:

**Figure supplement 1.** Murepavadin potentiates the activity of sub-lethal colistin concentrations, leading to enhanced CM damage, cell lysis and bacterial killing.

**Figure supplement 2.** Antibiogram summarising the antimicrobial susceptibilities of a panel of MDR *P. aeruginosa* human clinical strains isolated from sputum samples of cystic fibrosis patients.

killing (*Figure 5—figure supplement 2*). Of these 15 clinical isolates, 14 were susceptible to murepavadin alone, whilst one strain was resistant to the LptD inhibitor (*Supplementary file 1*). In 11 out of 14 murepavadin-susceptible CF isolates tested (79%), sub-lethal concentrations of murepavadin caused a significant increase in the bactericidal activity of colistin against *P. aeruginosa* (*Figure 5B*). Importantly, murepavadin did not affect the bactericidal activity of colistin against the strain that was resistant to the LptD inhibitor (*Figure 5B*). This confirmed that the potentiating effects of the LptD inhibitor on polymyxin-mediated killing were not due to off-target effects.

Next, we employed a high inoculum *P. aeruginosa* murine lung infection model and used a short treatment duration to assess how quickly combined colistin and murepavadin therapy could reduce bacterial burden, relative to mono-therapy. Mice were inoculated via the intranasal route with *P. aeruginosa* PA14 to cause a lung infection, and then treated intranasally with PBS alone, or PBS containing colistin only (5 mg kg$^{-1}$), murepavadin only (0.25 mg kg$^{-1}$), or colistin and murepavadin combined at the concentrations used for mono-treatment. These concentrations were based on those used previously to mimic treatment of human lung infections, and the route of delivery is similar to that used clinically (*Bernardini et al., 2019*; *Yapa et al., 2014*; *Melchers et al., 2019*; *Aoki et al., 2009*). Mono-therapy with colistin alone or murepavadin alone had very little effect on the bacterial load assessed after 3 hr treatment compared with the no-treatment control (*Figure 5C*). By contrast, combination therapy with colistin and murepavadin caused a ~500-fold reduction in c.f.u. counts relative to the no-treatment control (*Figure 5C*). Therefore, murepavadin synergises with colistin both in vitro and in vivo, suggesting it may be useful as a combination therapeutic approach for lung infections caused by *P. aeruginosa*.

## Discussion

Colistin is an increasingly important last-resort antibiotic used to treat infections caused by multi-drug-resistant Gram-negative pathogens, including *P. aeruginosa*, *K. pneumoniae*, and *E. coli* (*Garg et al., 2017*; *Biswas et al., 2012*; *Thomas et al., 2019*). However, treatment failure occurs frequently, and resistance is a growing concern (*Falagas et al., 2010*; *Paul et al., 2018*; *Linden et al., 2003*; *Tran et al., 2016*; *Satlin et al., 2020*; *MacNair et al., 2018*). Efforts to address these issues are compromised by a poor understanding of colistin's bactericidal mode of action. Whilst the initial interactions of colistin with LPS in the OM of Gram-negative bacteria were well established, it was unclear how the antibiotic traversed the OM and damaged the CM to cause cell lysis (*Figure 1—figure supplement 1*). In this work, we demonstrate that colistin targets LPS in the CM, resulting in membrane permeabilisation, bacterial lysis, and killing (*Figure 6*).

Our conclusion that colistin targets LPS in the CM was based initially on experiments with *E. coli* expressing the *mcr*-1 colistin resistance determinant. MCR-1 modifies lipid A with a pEtN moiety as it passes through the CM on its way to the OM (*Liu et al., 2016*; *Nang et al., 2019*). Since MCR-1 specifically protected spheroplasts from colistin but not nisin or daptomycin, which both target phospholipid bilayers, it was clear that colistin did not share the same target as the other two CAMPs. Given the only difference between *E. coli* spheroplasts expressing *mcr*-1 and pEmpty spheroplasts was modified LPS, our data reveal that colistin targets LPS in the CM, leading to disruption of the CM, which is a pre-requisite for subsequent cell lysis and bacterial killing (*Allison and*

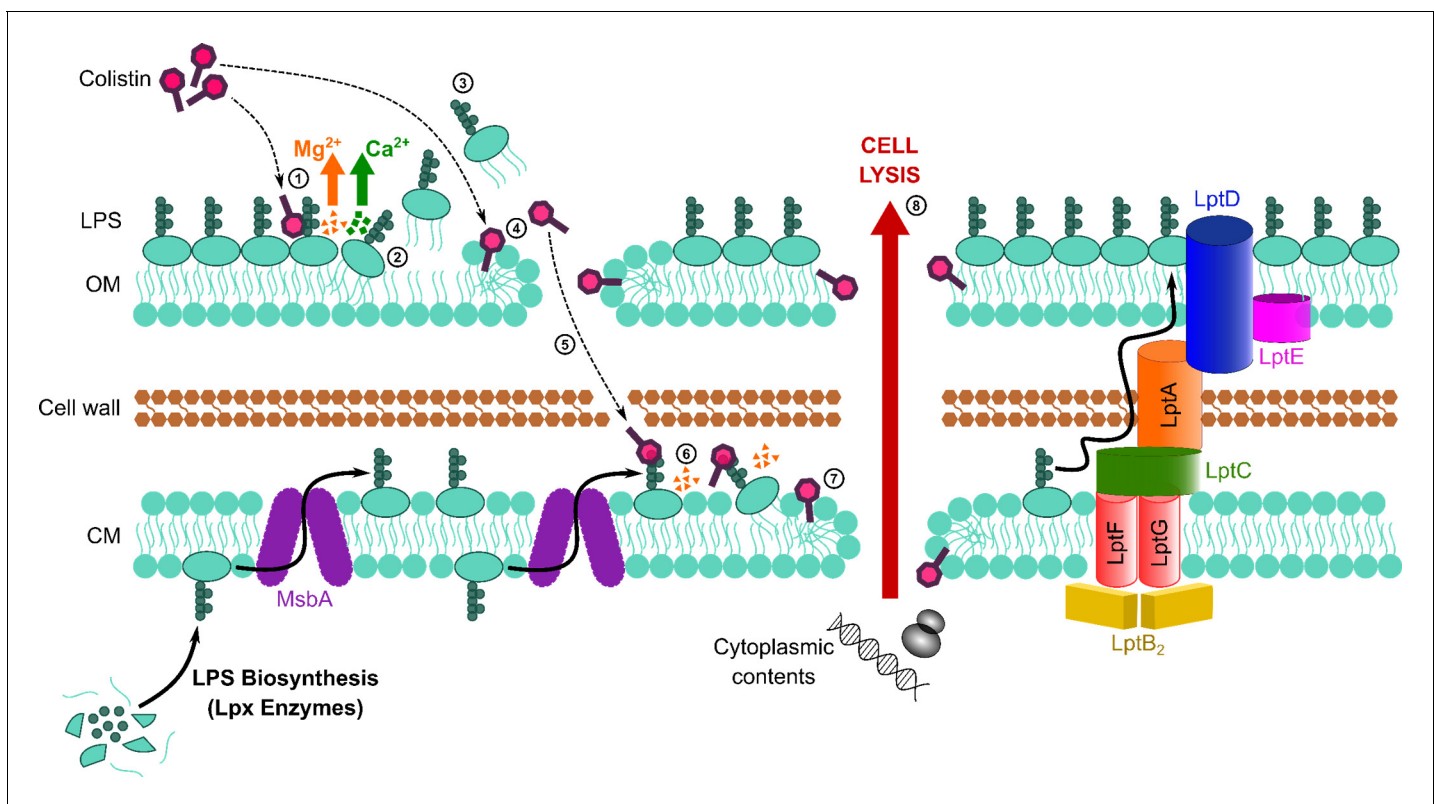

**Figure 6.** Colistin kills bacteria by targeting LPS in both the outer and cytoplasmic membrane (CM), leading to disruption of the cell envelope and bacterial lysis. Diagrammatic representation of the novel proposed mechanism of action of colistin: Colistin binds to LPS in the OM (1), displacing cations that form bridges between LPS molecules, which leads to destabilisation of the OM (2). As a consequence of the weakening of intermolecular bonds in the LPS monolayer, LPS is released from the bacterial surface (3), allowing colistin to further damage the OM via the action of the polymyxin lipid tail (4). This provides colistin with access to the periplasm, where colistin interacts with LPS in the CM (5) that is awaiting transport to the OM by the LptABCDEFG machinery after being synthesised in the cytoplasm and flipped to the outer leaflet of the CM by MsbA. As in the OM, colistin binding to LPS results in displacement of cation bridges and disruption of the CM (6), which it ultimately permeabilises (7), culminating in the loss of cytoplasmic contents, cell lysis, and bacterial death (8).

*Lambert, 2015*). These findings were then supported by experiments showing that LPS accumulation in the CM of *P. aeruginosa* sensitised this bacterium to colistin.

Similar to *Salmonella* and *E. coli*, the abundance of LPS in the CM of *P. aeruginosa* was found to be ~100 times lower than the OM, indicating that only about 1% of total LPS is present in the CM (*Zendo et al., 2010*; *Osborn et al., 1972*). However, studies with model membranes have shown that the presence of low concentrations of LPS (1% total composition) in phospholipid bilayer membranes was both necessary and sufficient for colistin-mediated permeabilisation (*Khadka et al., 2018*). Therefore, our conclusion explains how an antibiotic with a high degree of specificity for LPS could damage both the OM and CM (*Velkov et al., 2010*). The reason why MCR-1 protected the CM but not OM from colistin-mediated damage is most likely due to the lower proportion of unmodified LPS at the CM (21 ± 2%) relative to the OM (42 ± 19%) (*Figure 1A*). Furthermore, the overall abundance of LPS in the CM is very low, resulting in very few targets (i.e. unmodified LPS molecules) for colistin in the CM of *mcr-1*-expressing *E. coli*. By contrast, the OM of MCR-1-producing *E. coli* contains many more unmodified LPS molecules that can be bound by colistin, explaining why colistin is able to damage this structure, but cannot permeabilise the CM of *E. coli* expressing *mcr-1* at a physiologically relevant concentration (*Khadka et al., 2018*). Whether colistin resistance conferred by chromosomal mutations in two-component systems is also mediated by modified LPS in the CM remains to be tested (*Carroll et al., 2019*; *Nang et al., 2019*; *Skov and Monnet, 2016*; *Wang et al., 2020*; *Raetz et al., 2007*; *Simpson and Trent, 2019*).

Our data showing that colistin requires unmodified LPS to be present in the CM to kill bacteria explains how an antibiotic with high affinity and specificity for LPS causes disruption to both the OM and CM (*Velkov et al., 2010*). Furthermore, these findings provide support for the observations that colistin does not damage the OM of colistin-resistant *A. baumannii* isolates where LPS has been replaced by a phospholipid bilayer, and that polymyxins cause only minimal disruption to model phospholipid membranes unless LPS is present (*Khadka et al., 2018*; *Moffatt et al., 2010*; *Zhang et al., 2018*; *Fu et al., 2020*).

Whilst the interaction of colistin with LPS in the CM is likely to share similarities with the same process at the OM, there are also likely to be differences owing to the differing concentrations of LPS between the two membranes (*Khadka et al., 2018*; *Zendo et al., 2010*; *Osborn et al., 1972*). In the OM, LPS is a highly abundant component with molecules tightly packed and stabilised with cation bridges. By contrast, LPS is a minority component in the CM, which may affect the rate and degree to which the CM is disrupted by polymyxins. In support of this, whilst colistin induced OM damage within minutes of bacterial exposure to the antibiotic, disruption of the CM took much longer. Even when spheroplasts lacking an OM were exposed to colistin, it still took more than 2 hr for CM permeabilisation to occur. Therefore, it appears that colistin-mediated disruption of the CM is considerably less efficient than that of the OM, likely due to the much lower levels of LPS present in the CM.

In addition to disruption of both the OM and IM, it has been proposed that the lethal activity of polymyxin antibiotics may be due, at least in part, to: disruption of NADH-quinone reductase; the generation of reactive oxygen species (ROS); the binding of the antibiotic to ribosomes; and the fusion of the OM and CM, leading to phospholipid exchange (*Cajal et al., 1996*; *Clausell et al., 2006*; *Clausell et al., 2007*; *Deris et al., 2014*; *Ajiboye et al., 2018*; *Li and Velkov, 2019*; *Ayoub Moubareck, 2020*; *El-Sayed Ahmed et al., 2020*). However, whilst these have been considered as discrete events or alternative mechanisms of action, it is possible that all these phenomena occur as downstream consequences of colistin-mediated CM disruption. For example, NADH-quinone reductase is a component of the electronic transport chain (ETC), which is located within the CM and may therefore be disrupted by membrane damage, whilst the generation of ROS may arise via disruption of the ETC as has been proposed for the CAMP LL-37 (*Deris et al., 2014*; *Ayoub Moubareck, 2020*; *Choi et al., 2017*). The fusion of the OM and CM and subsequent exchange of lipids appears to depend upon the interaction of the polymyxin with, and presumably disruption of, both membranes (*Cajal et al., 1996*; *Clausell et al., 2006*; *Clausell et al., 2007*; *Ayoub Moubareck, 2020*; *El-Sayed Ahmed et al., 2020*). Finally, the interaction of polymyxins with ribosomes requires the antibiotic to pass through the CM to access the cytoplasm (*McCoy et al., 2013*). Therefore, whilst the disruption of the CM by polymyxin antibiotics is the key step required for bacterial killing, this may be due to multiple deleterious effects on cellular processes.

Our findings provide strong evidence that colistin targets LPS in the CM, in addition to the OM, and that this is required for the bactericidal and lytic activity of the antibiotic at clinically relevant concentrations. This insight into the mode of action of colistin enabled us to devise a new therapeutic approach to enhance colistin efficacy. Using the LptD inhibitor murepavadin, which is in development as an inhaled treatment for *P. aeruginosa* infections, we triggered LPS accumulation in the CM, and thereby increased the susceptibility of bacteria to colistin (*Lehman and Grabowicz, 2019*). The potential clinical utility of this approach was demonstrated by showing enhanced activity of colistin–murepavadin combination therapy against a panel of clinical CF isolates, as well as potent efficacy in a murine model of *P. aeruginosa* lung infection. It is anticipated that a combination of colistin and murepavadin could enhance the low treatment efficacy of polymyxin antibiotics and may also limit the toxic side effects associated with both compounds by enabling the use of lower doses of the drugs (*Lehman and Grabowicz, 2019*).

Interestingly, whilst we found that blocking LPS transport to the OM sensitised bacteria to colistin, previous work has shown that novobiocin increases the susceptibility of bacteria to colistin by increasing LPS transport to the OM (*Mandler et al., 2018*). This might suggest that novobiocin reduces LPS levels in the CM and thus contradicts our findings. However, transport of LPS to the OM is regulated such that this process does not deplete LPS in the CM (*Xie et al., 2018*). Furthermore, LPS biosynthesis is tightly regulated at the CM in response to the abundance of LPS by PbgA, LapB, and FtsH (*Clairfeuille et al., 2020*; *Guest et al., 2020*; *Fivenson and Bernhardt, 2020*; *O'Rourke et al., 2020*; *Lee et al., 2006*). Therefore, it is expected that LPS synthesis would be increased to address the elevated rate of transport to the OM, which might lead to elevated levels of LPS in the CM as it is produced to meet demand. In support of this, novobiocin exposure increases the expression of *lpxC* in *E. coli*, which encodes the enzyme that is the first committed step in LPS biosynthesis and provides a key checkpoint in LPS production (*Raetz et al., 2007*; *Simpson and Trent, 2019*; *O'Rourke et al., 2020*). However, the effect of novobiocin on LPS abundance in the CM remains to be tested.

It should be noted that whilst bacteria can modulate LPS biosynthesis to maintain LPS abundance in the CM, there is no mechanism to remove LPS from the CM, which is why LPS accumulation occurs with murepavadin.

In summary, this work contributes to our understanding of the mechanism of action of colistin by demonstrating that polymyxin antibiotics target LPS in both the OM and the CM, and that this leads to the disruption of both membranes, resulting in the bactericidal and lytic activities of the antibiotic. Modulation of LPS levels in the CM can enhance colistin activity, providing the foundations for new approaches to enhance the efficacy of this antibiotic of last resort.

# Materials and methods

### Key resources table

| Reagent type (species) or resource | Designation | Source or reference | Identifiers | Additional information |
|---|---|---|---|---|
| Strain, strain background (*Escherichia coli*) | MC1000 | *Dortet et al., 2018* PMID:30442963 | pEmpty | Background strain (*araD139*, Δ(ara, leu)7697, Δ*lacX74, galU, galK, strA*) harbouring the IPTG-inducible pDM1 plasmid (GenBank MN128719) |
| Strain, strain background (*Escherichia coli*) | MC1000 | *Dortet et al., 2018* PMID:30442963 | *mcr*-1 | MC1000 strain harbouring the pDM1 plasmid encoding the *mcr-1* gene amplified from a clinical *E. coli* isolate |
| Strain, strain background (*Pseudomonas aeruginosa*) | PA14 | *Lee et al., 2006* PMID:17038190 | PA14 | Wild-type reference strain; highly virulent human isolate representing most common clonal group worldwide |

*Continued on next page*

Continued

| Reagent type (species) or resource | Designation | Source or reference | Identifiers | Additional information |
|---|---|---|---|---|
| Strain, strain background (*Pseudomonas aeruginosa*) | AK3 | This study | AK3 | Multi-drug resistant human clinical isolate from sputum of cystic fibrosis patient – mucoid strain |
| Strain, strain background (*Pseudomonas aeruginosa*) | AK10 | This study | AK10 | Multi-drug resistant human clinical isolate from sputum of cystic fibrosis patient |
| Strain, strain background (*Pseudomonas aeruginosa*) | AK20 | This study | AK20 | Multi-drug resistant human clinical isolate from sputum of cystic fibrosis patient |
| Strain, strain background (*Pseudomonas aeruginosa*) | AK6 | This study | AK6 | Multi-drug resistant human clinical isolate from sputum of cystic fibrosis patient – mucoid strain |
| Strain, strain background (*Pseudomonas aeruginosa*) | AK12 | This study | AK12 | Multi-drug resistant human clinical isolate from sputum of cystic fibrosis patient – mucoid strain |
| Strain, strain background (*Pseudomonas aeruginosa*) | AK8 | This study | AK8 | Multi-drug resistant human clinical isolate from sputum of cystic fibrosis patient – mucoid strain |
| Strain, strain background (*Pseudomonas aeruginosa*) | AK17 | This study | AK17 | Multi-drug resistant human clinical isolate from sputum of cystic fibrosis patient |
| Strain, strain background (*Pseudomonas aeruginosa*) | AK9 | This study | AK9 | Multi-drug resistant human clinical isolate from sputum of cystic fibrosis patient |
| Strain, strain background (*Pseudomonas aeruginosa*) | AK14 | This study | AK14 | Multi-drug resistant human clinical isolate from sputum of cystic fibrosis patient |
| Strain, strain background (*Pseudomonas aeruginosa*) | AK5 | This study | AK5 | Multi-drug resistant human clinical isolate from sputum of cystic fibrosis patient – mucoid strain |
| Strain, strain background (*Pseudomonas aeruginosa*) | AK11 | This study | AK11 | Multi-drug resistant human clinical isolate from sputum of cystic fibrosis patient – mucoid strain |
| Strain, strain background (*Pseudomonas aeruginosa*) | AK13 | This study | AK13 | Multi-drug resistant human clinical isolate from sputum of cystic fibrosis patient |
| Strain, strain background (*Pseudomonas aeruginosa*) | AK18 | This study | AK18 | Multi-drug resistant human clinical isolate from sputum of cystic fibrosis patient |
| Strain, strain background (*Pseudomonas aeruginosa*) | AK22 | This study | AK22 | Multi-drug resistant human clinical isolate from sputum of cystic fibrosis patient |
| Strain, strain background (*Pseudomonas aeruginosa*) | AK19 | This study | AK19 | Multi-drug resistant human clinical isolate from sputum of cystic fibrosis patient – murepavadin-resistant |
| Chemical compound, drug | Colistin | Sigma-Aldrich | C4461-1G | Targets LPS |
| Chemical compound, drug | Murepavadin (POL7080) | DC Chemicals | DC11273 | Targets LptD |
| Chemical compound, drug | Daptomycin | Bio-Techne Ltd | 3917/10 | Targets phosphatidylglycerol in the bacterial membrane |

*Continued on next page*

*Continued*

| Reagent type (species) or resource | Designation | Source or reference | Identifiers | Additional information |
|---|---|---|---|---|
| Chemical compound, drug | Nisin | Sigma-Aldrich | N5764-5G | Targets bacterial membranes |
| Chemical compound, drug | Rifampicin | Molekula Ltd | 32609202 | Targets RNA Polymerase |
| Chemical compound, drug | Polymyxin B nonapeptide (PMBN) | Sigma-Aldrich | P2076-5MG | Permeabilises the OM |
| Chemical compound, drug | Tetracycline | Sigma-Aldrich | 87128–25G | Protein synthesis inhibitor |
| Commercial assay or kit | LAL Chromogenic Endotoxin Quantitation Kit | Thermo Scientific Pierce | 88282 | Quantitative assay for LPS |
| Chemical compound, drug | Isopropyl β-d-1-thiogalactopyranoside (IPTG) | Melford Laboratories | MB1008 | Induces gene expression |
| Chemical compound, drug | Lysozyme from chicken egg white | Sigma-Aldrich | L6876-1G | Degrades peptidoglycan |
| Chemical compound, drug | | | | |

Ethylenediaminetetraacetic acid (EDTA)Sigma-AldrichE6511-100GRemoves OM from bacteriaChemical compound, drugTrypsinSigma-AldrichT7309-1GDegrades proteinsChemical compound, drugPropidium iodide (PI)Sigma-AldrichP4864-10MLFluoresces when bound to DNAChemical compound, drugFluorescein 5 (6)-isothiocyanateSigma-Aldrich46950–50 MG-FUsed to label proteins with a fluorescent tagChemical compound, drugN-phenyl-1-napthylamine (NPN)Acros Organics147160250Fluoresces when bound to phospholipidsChemical compound, drugFITC-labelled Poly-L-lysine (PLL)Sigma-AldrichP3543-10MGUsed to measure membrane surface chargeChemical compound, drug6-Dodecanoyl-N,N-dimethyl-2-naphthylamine (Laurdan)Sigma-Aldrich40227–100 MGUsed to measure membrane fluidity

## Bacterial strains and growth conditions

The bacterial strains used in this study are listed in Key resources table. For each experiment, all strains were grown in Luria broth (LB; Thermo Fisher Scientific, USA) for 18 hr to stationary phase at 37°C with shaking (180 r.p.m.). For routine culture of bacteria on solid media, strains were grown on LB supplemented with 1.5% technical agar (BD Biosciences, USA). Liquid and solid growth media were supplemented with tetracycline (12.5 µg ml$^{-1}$; Sigma-Aldrich, USA) and isopropyl-β-D-thiogalactoside (IPTG, Melford Laboratories, UK; used at 0.5 mM unless stated otherwise) where required. Enumeration of bacterial c.f.u. was done by plating 10-fold serial dilutions of bacterial cultures on to Mueller-Hinton agar (MHA; Thermo Fisher Scientific) plates. Inoculated agar plates were incubated statically for 18 hr in air at 37°C.

## Determination of MICs of antibiotics

The MIC of colistin and murepavadin against bacterial strains was determined by the broth microdilution protocol (*Wiegand et al., 2008*). A microtitre plate was used to prepare a range of antibiotic concentrations in 200 µl cation-adjusted Mueller-Hinton broth (CA-MHB; Thermo Fisher Scientific) by two-fold serial dilutions. For certain experiments, checkerboard analyses were performed by preparing two-fold serial dilutions of two antibiotics in different directions, generating an 8 × 8 matrix to assess the MICs of the relevant antibiotics in combination, with FICI values calculated as previously described (*Odds, 2003*). Stationary-phase bacteria were diluted 1000-fold in fresh CA-MHB and seeded into each well of the microtitre plate to a final concentration of 5 × 10$^5$ c.f.u. ml$^{-1}$. The microtitre plates were then incubated statically at 37°C for 18 hr in air, after which point the MIC was defined as the lowest antibiotic concentration at which there was no visible growth of bacteria. In some cases, the extent of bacterial growth after 18 hr incubation was also determined by obtaining OD$_{595nm}$ measurements using a Bio-Rad iMark microplate absorbance reader (Bio-Rad Laboratories, USA).

## Bacterial growth assay

Stationary-phase bacteria were diluted 1000-fold in fresh CA-MHB, and 4 µl was seeded into the wells of a microtitre plate containing 200 µl CA-MHB, and for some experiments the LptD inhibitor murepavadin, to give a final inoculum of 5 × 10$^5$ c.f.u. ml$^{-1}$. The microtitre plate was incubated with

shaking (180 r.p.m.) at 37°C for 16 hr in a Tecan Infinite 200 Pro multiwell plate reader (Tecan Group Ltd., Switzerland) and optical density measurements were taken at 600 nm every 15 min.

## Production of spheroplasts

Spheroplasts of *E. coli* and *P. aeruginosa* strains lacking an OM and cell wall were generated as previously described (*Weiss and Fraser, 1973*). Briefly, stationary-phase bacteria grown overnight were washed twice by centrifugation (12,300 × *g*, 3 min) followed by resuspension in CA-MHB, and added at a final inoculum of $10^8$ c.f.u. ml$^{-1}$ to 9 ml CA-MHB containing for some experiments varying concentration of IPTG, or the LPS transport inhibitor murepavadin. Cultures were then incubated at 37°C with shaking (180 r.p.m.) for 2 hr. After the incubation, bacteria were washed twice by centrifuging (3273 × *g*, 20 min, 4°C) and resuspending first in 10 ml Tris buffer (0.03 M, pH 8.0; Sigma-Aldrich), and subsequently in Tris buffer (0.03 M, pH 8.0) containing 20% sucrose. EDTA (250 µl, 10 mg ml$^{-1}$; Sigma-Aldrich) and lysozyme (1 ml, 10 mg ml$^{-1}$; Roche, Switzerland) were added to remove the OM and cell wall respectively, and the cell suspension was incubated for 1 hr in a water bath shaker at 30°C. Trypsin (500 µl, 10 mg ml$^{-1}$; Sigma-Aldrich) was then added, and the culture again incubated at 30°C in a water bath shaker for 15 min. The resulting spheroplasts produced were harvested by mild centrifugation (2000 × *g*, 20 min, 4°C), with the supernatant containing the removed OM extracted for further analysis. Successful conversion of bacterial whole cells into spheroplasts was confirmed using phase-contrast microscopy, as detailed below.

## Confirmation of successful spheroplast formation

Whole cells of *E. coli* and *P. aeruginosa* grown overnight were washed twice by centrifugation (12,300 × *g*, 3 min) and resuspension in CA-MHB, added at a final inoculum of $10^8$ c.f.u. ml$^{-1}$ to 9 ml CA-MHB, and incubated for 2 hr at 37°C with shaking (180 r.p.m.). OM proteins of these bacteria were subsequently labelled with fluorescein isothiocyanate (FITC, Sigma-Aldrich) as previously described (*Loh and Ward, 2012*). Bacterial cells were washed twice by centrifugation (3273 × *g*, 20 min, 4°C) and resuspension in 10 ml Labelling Buffer (50 mM Na$_2$CO$_3$, 100 mM NaCl, pH 8.0), to which FITC was added at a final concentration of 0.5 mg ml$^{-1}$. Bacteria were incubated for 30 min at room temperature, before labelled cells were harvested by centrifuging (3273 × *g*, 20 min, 4°C) and washed thrice by resuspending in 10 ml Tris buffer (0.03 M, pH 8.0) containing 20% sucrose. 1 ml of FITC-labelled bacteria was extracted and centrifuged (12,300 × *g*, 3 min), and the cells were fixed in 4% paraformaldehyde (PFA) in phosphate-buffered saline (PBS). The remaining 9 ml of FITC-labelled cells were converted into spheroplasts, as described above. The spheroplasts produced were recovered by mild centrifugation (2000 × *g*, 20 min, 4°C) and resuspension in 9 ml Tris buffer (0.03 M, pH 8.0) containing 20% sucrose, before 1 ml of spheroplasts were fixed in the same way as with whole cells. The amount of FITC fluorescence in the OM of whole cells and CM of spheroplasts was observed using fluorescence microscopy, as described below. For quantification of FITC fluorescence, 200 µl samples of the fixed bacterial suspensions were seeded into the wells of a black-walled microtitre plate, and fluorescence measured with a Tecan Infinite 200 Pro multiwell plate reader, using an excitation wavelength of 490 nm and an emission wavelength of 525 nm.

## Microscopy

For phase-contrast and fluorescence microscopy, a 5 µl sample of fixed bacterial whole cells or spheroplasts was spotted onto a thin 1.2% agarose gel patch prepared in distilled water on a microscope slide. Bacteria were imaged using an Axio Imager.A2 Zeiss microscope (Carl Zeiss Microscopy GmbH, Germany) at 1000× magnification with an oil immersion objective lens. The ZEN 2012 software was used for image acquisition, whilst analysis of cell length:width ratios was done using the FIJI/ImageJ software by measuring two perpendicular lines drawn through the centre of bacteria. For each experiment, all microscopy images were acquired and processed using identical settings throughout.

## Determination of LPS concentration and modification by mass spectroscopy

Spheroplasts from bacterial cells were prepared as described above and then resuspended in ddH$_2$O (200 µl), before mild acid hydrolysis was performed via the addition of 2% (vol/vol) acetic

acid in ddH$_2$O (200 µl) and incubation at 100°C for 30 min. For experiments with whole cells, bacteria grown overnight to stationary-phase were washed three times by centrifuging and resuspending in ddH$_2$O, and a mild acid hydrolysis was performed on these whole cells as described for spheroplasts. Acid-treated whole cells or spheroplasts were recovered by centrifugation (17,000 × *g*, 2 min), and the resulting pellet was washed before being resuspended in 50 µl ultrapure water. The whole cell or spheroplast suspension (0.5 µl) was then loaded immediately onto the target and overlaid with 1.2 µl of a matrix consisting of 9H-Pyrido[3,4-B]indole (Norharmane) (Sigma-Aldrich) dissolved in 90:10 (vol/vol) chloroform/methanol to a final concentration of 10 mg ml$^{-1}$. The bacterial suspension and matrix were then mixed on the target before gentle drying under air at room temperature. MALDI-TOF mass spectroscopy analysis was undertaken with a MALDI Biotyper Sirius system (Bruker Daltonics, USA), using the linear negative-ion mode as described previously (*Furniss et al., 2019*). Manual peak picking at masses relevant to phospholipids or lipid A was performed on the mass spectra obtained, and the corresponding signal intensities at the defined masses were determined. Peaks were considered only if their signal/noise ratio was at least 5. To determine the ratio of modified lipid A to unmodified lipid A, the area under the pETN-modified lipid A peak (*m/z* 1,919.2) was divided by the area under the peak corresponding to native lipid A (*m/z* 1,796.2). To determine the relative abundance of LPS, the sum of the area under the lipid A peaks (*m/z* 1447–1700) was divided by the sum of the area under representative phospholipid peaks (phospholipid 34:1,2, *m/z* 717–747). All mass spectra were generated and analysed with three biological replicates and two technical replicates.

## OM disruption assay

To detect damage to the OM of bacteria, the well-established NPN uptake assay was used (*Helander and Mattila-Sandholm, 2000*). Stationary-phase bacterial cells were washed in fresh CA-MHB and diluted to an optical density (OD$_{600nm}$) of 0.5 in 5 mM pH 7.2 HEPES buffer (Sigma-Aldrich). This bacterial suspension was added to wells containing the relevant antibiotics in HEPES buffer, as well as the fluorescent probe *N*-phenyl-1-naphthylamine (NPN; Acros Organics, USA) at a final concentration of 10 µM. Samples were placed in a black microtitre plate with clear-bottomed wells and fluorescence measured immediately in a Tecan Infinite 200 Pro multiwell plate using an excitation wavelength of 355 nm and an emission wavelength of 405 nm. Fluorescence measurements were obtained every 30 s for 10 min, and the degree of OM permeabilisation, referred to as the NPN Uptake Factor, was calculated using the following formula:

$$\frac{\text{Fluorescence of sample with NPN} - \text{Fluorescence of sample without NPN}}{\text{Fluorescence of HEPES buffer with NPN} - \text{Fluorescence of HEPES buffer without NPN}}$$

## CM disruption assay

To measure CM disruption of whole cells, bacteria grown to stationary-phase overnight were washed and inoculated into 3 ml MHB containing the relevant antibiotics. Cultures were incubated at 37°C with shaking (180 r.p.m.) for up to 8 hr, and every 30 min, aliquots (200 µl) were taken and bacteria isolated by centrifugation (12,300 × *g*, 3 min). Cells were then washed in sterile PBS before being added to the wells of a black-walled microtitre plate, and PI (Sigma-Aldrich) was added to each well at a final concentration of 2.5 µM. Fluorescence was measured immediately in a Tecan Infinite 200 Pro multiwell plate reader (excitation at 535 nm, emission at 617 nm). To measure disruption of the CM in spheroplasts, spheroplasts of *E. coli* and *P. aeruginosa* generated as detailed above were washed by centrifugation (4000 × *g*, 5 min) and resuspension in Tris buffer (0.03 M, pH 8.0) containing 20% sucrose. Spheroplast samples (20 µl) were then added in the wells of a black-walled microtitre plate to 180 µl of Tris buffer (0.03 M, pH 8.0) containing 20% sucrose, the relevant antibiotics, and PI at a final concentration of 0.25 µM. The microtitre plate was incubated with shaking (180 r.p.m.) at 37°C for up to 8 hr in a Tecan Infinite 200 Pro multiwell plate reader and fluorescence (excitation at 535 nm, emission at 617 nm) measured every 15 min using a gain of 80 or 100. For both whole bacterial cells and spheroplasts, to account for differences in fluorescence values arising from variations in cell number, relative fluorescence unit (r.f.u.) measurements were corrected for OD at 600 nm.

## Determination of bacterial lysis

In the case of whole bacterial cells, washed stationary-phase bacteria were inoculated into 3 ml CA-MHB containing the relevant antibiotics, as described above. Cultures were then placed in a shaking incubator (37°C, 180 r.p.m.) for 8 hr, and every 30 min, samples (200 μl) were transferred to a microtitre plate, where $OD_{595nm}$ measurements were obtained using a Bio-Rad iMark microplate absorbance reader. For spheroplasts, washed spheroplasts (20 μl) were added to 180 μl of Tris buffer (0.03 M, pH 8.0) containing 20% sucrose and the relevant antibiotics in a microtitre plate as detailed above. The microtitre plate was incubated for up to 8 hr at 37°C with shaking (180 r.p.m.) in a Tecan Infinite 200 Pro multiwell plate reader, and readings of $OD_{600nm}$ were made every 15 min.

## Determination of membrane fluidity

The fluidity of the CM of spheroplasts was assessed using the fluorescent dye Laurdan, as previously described (*Müller et al., 2016*). Washed spheroplasts of *E. coli* (500 μl) prepared as described above were incubated at room temperature for 5 min in Tris buffer (0.03 M, pH 8.0) containing 20% sucrose and Laurdan at a final concentration of 100 μM. Spheroplast samples were washed by three rounds of centrifugation (4000 × *g*, 5 min) and resuspension in Tris buffer containing 20% sucrose, then 200 μl was transferred to the wells of a black-walled microtitre plate. Membrane fluidity was measured using a Tecan Infinite 200 Pro multiwell plate reader, with fluorescence determined using an excitation wavelength of 330 nm, and emission wavelengths of 460 nm and 500 nm. Generalised Polarisation (GP) values were calculated using the following formula:

$$GP = \frac{\text{Emission intensity at } 460\,nm - \text{Emission intensity at } 500\,nm}{\text{Emission intensity at } 460\,nm + \text{Emission intensity at } 500\,nm}$$

A higher GP value indicated a membrane with reduced fluidity, with altered water penetration into the membrane affecting the fluorescence of the Laurdan dye.

## Determination of membrane charge

The charge of the CM of spheroplasts was measured using FITC-labelled Poly-L-Lysine (PLL) (*Jones et al., 2008*). Washed *E. coli* spheroplasts (300 μl) generated as described above were incubated in the dark for 10 min at room temperature in Tris buffer (0.03 M, pH 8.0) containing 20% sucrose and FITC-PLL at a final concentration of 20 μg ml$^{-1}$. To remove any unbound PLL, spheroplasts were subsequently washed thoroughly by three rounds of centrifugation (4000 × *g*, 5 min) and resuspension in Tris buffer containing 20% sucrose. Spheroplast samples (200 μl) were seeded into the wells of a black-walled microtitre plate, and FITC fluorescence was quantified in a Tecan Infinite 200 Pro multiwell plate reader (excitation at 490 nm, emission at 525 nm). Reduced PLL binding to the surface of spheroplasts indicated a more positively charged membrane, with the cationic FITC fluorophore having less affinity for the CM.

## Determination of LPS concentration by Limulus Amebocyte Lysate assay

Stationary-phase bacteria grown overnight were washed and grown for 2 hr, before conversion to spheroplasts as described above. During formation of spheroplasts, the OM extracted from the bacterial cells was recovered, and the concentration of LPS in the OM, as well as the concentration of LPS in the CM of spheroplasts, was determined using the chromogenic Limulus Amebocyte Lysate (LAL) assay (all reagents from Thermo Fisher Scientific) as described previously (*Lam et al., 2014*). OM samples and spheroplasts lysed by freeze-thaw to release LPS were diluted in 10-fold steps, and 50 μl of each sample was equilibrated to 37°C and loaded into the wells of a microtitre plate at the same temperature. LAL reagent (50 μl) was added to each well, and the mixture incubated at 37°C for 10 min. Chromogenic substrate solution (100 μl, 2 mM) was subsequently added to each well and the microtitre plate was incubated for a further 6 min at 37°C. The enzymatic reaction was stopped by adding 50 μl of 25% acetic acid to each well, and the presence of LPS was determined by measuring absorbance at 405 nm in a Tecan Infinite 200 Pro multiwell plate reader. A standard curve was generated using an *E. coli* endotoxin standard stock solution, which enabled the conversion of $A_{405nm}$ values into concentrations of LPS.

## Determination of bactericidal activity of antibiotics

As described above, stationary-phase bacteria grown overnight were washed twice and added at a final inoculum of $10^8$ c.f.u. $ml^{-1}$ to 3 ml CA-MHB containing colistin and/or murepavadin. Cultures were incubated with shaking (37°C, 180 r.p.m.) for up to 8 hr. Bacterial survival was determined after 2, 4, 6, and 8 hr by serially diluting cultures in 10-fold steps in 200 µl sterile PBS (VWR International, USA), before enumeration of c.f.u. counts on MHA plates.

## Murine lung infection model

The use of mice was performed under the authority of the UK Home Office outlined in the Animals (Scientific Procedures) Act 1986 after ethical review by Imperial College London Animal Welfare and Ethical Review Body (PPL 70/7969). Wild-type C57BL/6 mice were purchased from Charles River (UK). All mice were female and aged between 6 and 8 weeks. Mice were housed with five per cage with Aspen chip 2 bedding and 12 h light/dark cycles at 20–22°C. Mice were randomly assigned to experimental groups. Water was provided ad libitum and mice were fed RM1 (Special Diet Services). To establish colonisation of the lungs, mice were anesthetised and intranasally inoculated with $10^7$ c. f.u. of *P. aeruginosa* PA14 in 50 µl of PBS, as described previously (*Clarke, 2014*; *Brown et al., 2017*). Infection was allowed to establish for 5 hr, before mice were again anaesthetised and treated via the intranasal route with 50 µl of PBS alone, or PBS containing colistin (5 mg $kg^{-1}$), murepavadin (0.25 mg $kg^{-1}$), or a combination of colistin and murepavadin for 3 hr. To enumerate bacterial load in the lungs, mice were humanely sacrificed, their lungs removed and homogenised in PBS, and then plated onto Pseudomonas isolation agar (Thermo Fisher Scientific).

## Statistical analyses

Experiments were performed on at least three independent occasions, and the resulting data are presented as the arithmetic mean of these biological repeats, unless stated otherwise. Error bars, where shown, represent the standard deviation of the mean. For single comparisons, a two-tailed Student's *t*-test was used to analyse the data. For multiple comparisons at a single time point or concentration, data were analysed using a one-way analysis of variance (ANOVA) or a Kruskal–Wallis test. Where data were obtained at several different time points or concentrations, a two-way ANOVA was used for statistical analyses. Appropriate post hoc tests (Dunnett's, Tukey's, Sidak's, Dunn's) were carried out to correct for multiple comparisons, with details provided in the figure legends. Asterisks on graphs indicate significant differences between data, and the corresponding p-values are reported in the figure legend. All statistical analyses were performed using GraphPad Prism 7 software (GraphPad Software Inc, USA).

## Acknowledgements

Aishwarya Krishna, Laura Nolan, Alain Filloux, Ollie Fletcher (all Imperial College London), and Alireza Abdolrasouli (Imperial College Healthcare NHS Trust) are gratefully acknowledged for providing bacterial strains. Lisa Haigh (Imperial College London) is thanked for processing and analysing mass spectrometry samples. Vladimir Pelicic (Imperial College London) is thanked for providing access to the Axio Imager.A2 Zeiss microscope. AS is supported by a PhD studentship funded by a Medical Research Council Doctoral Training Award to Imperial College London (MR/N014103/1). AK is funded by the DFG German Research Foundation (KL3191/1-1) and the European Union's Horizon 2020 research and innovation programme under the Marie Skłodowska-Curie Actions grant agreement 'BacDrug' (838183). MB and MMS gratefully acknowledge support from the Rosetrees Trust (M300-CD1). DAIM and RCDF gratefully acknowledge funding from an MRC Career Development Award (MR/M009505/1). JCD is supported by the NIHR through a Senior Investigator award and the Imperial Biomedical Research Centre (BRC). RM and JCD are supported by Cystic Fibrosis Trust funding. GJL-M is funded by the MRC Confidence in Concept Fund and a ISSF Wellcome Trust Grant (105603/Z/14/Z). TBC is a Sir Henry Dale Fellow jointly funded by the Wellcome Trust and Royal Society (107660/Z/15Z). LEE and AME acknowledge funding from the Wellcome Trust (204337/Z/16/Z). AME also acknowledges support from the National Institute for Health Research (NIHR) Imperial Biomedical Research Centre (BRC) and declares funding from Shionogi and Co., Ltd for an unrelated project.

## Additional information

### Funding

| Funder | Grant reference number | Author |
| --- | --- | --- |
| Medical Research Council | PhD Studentship | Akshay Sabnis |
| Deutsche Forschungsge-meinschaft | KL3191/1 | Anna Klöckner |
| Horizon 2020 | 838183 | Anna Klöckner |
| Rosetrees Trust | M300-CD1 | Michele Becce<br>Molly M Stevens |
| Medical Research Council | (MR/M009505/1) | R Christopher D Furniss<br>Despoina AI Mavridou |
| Cystic Fibrosis Trust | | Ronan Murphy<br>Jane C Davies |
| Wellcome Trust | 105603/Z/14/Z | Gérald J Larrouy-Maumus |
| Wellcome Trust | 107660/Z/15Z | Thomas B Clarke |
| Wellcome Trust | 204337/Z/16/Z | Lindsay E Evans<br>Andrew M Edwards |
| NIHR Biomedical Research Centre (Imperial College) | | Jane C Davies<br>Andrew M Edwards |

The funders had no role in study design, data collection and interpretation, or the decision to submit the work for publication.

### Author contributions

Akshay Sabnis, Conceptualization, Data curation, Formal analysis, Investigation, Methodology, Writing - original draft, Writing - review and editing; Katheryn LH Hagart, Michele Becce, Investigation, Writing - review and editing; Anna Klöckner, Formal analysis, Investigation; Lindsay E Evans, Formal analysis, Investigation, Writing - review and editing; R Christopher D Furniss, Resources, Investigation, Methodology, Writing - review and editing; Despoina AI Mavridou, Molly M Stevens, Resources, Supervision, Methodology, Writing - review and editing; Ronan Murphy, Resources, Methodology, Writing - review and editing; Jane C Davies, Resources, Writing - review and editing; Gérald J Larrouy-Maumus, Formal analysis, Supervision, Visualization, Methodology, Writing - review and editing; Thomas B Clarke, Data curation, Formal analysis, Investigation, Writing - review and editing; Andrew M Edwards, Conceptualization, Formal analysis, Supervision, Methodology, Writing - original draft, Project administration, Writing - review and editing

### Author ORCIDs

R Christopher D Furniss https://orcid.org/0000-0002-5806-5099
Molly M Stevens https://orcid.org/0000-0002-7335-266X
Andrew M Edwards https://orcid.org/0000-0002-7173-7355

### Ethics

Animal experimentation: The use of mice was performed under the authority of the UK Home Office outlined in the Animals (Scientific Procedures) Act 1986 after ethical review by Imperial College London Animal Welfare and Ethical Review Body (PPL 70/7969).

### Decision letter and Author response

Decision letter https://doi.org/10.7554/eLife.65836.sa1
Author response https://doi.org/10.7554/eLife.65836.sa2

## Additional files

### Supplementary files

- Supplementary file 1. Contains Supplementary Table 1 detailing the minimum inhibitory concentrations (MICs) of colistin and murepavadin against the *P. aeruginosa* strains used in this study.

- Transparent reporting form

### Data availability

Source data for all figures has been deposited at Dryad: https://doi.org/10.5061/dryad.98sf7m0hh.

The following dataset was generated:

| Author(s) | Year | Dataset title | Dataset URL | Database and Identifier |
|---|---|---|---|---|
| Sabnis A, Hagart KLH, Klöckner A, Becce M, Evans LE, Furniss RCD, Mavridou DAi, Murphy R, Stevens MM, Davies JC, Larrouy-Maumus GJ, Clarke TB, Edwards AM | 2021 | Data from: Colistin kills bacteria by targeting lipopolysaccharide in the cytoplasmic membrane | http://dx.doi.org/10.5061/dryad.98sf7m0hh | Dryad Digital Repository, 10.5061/dryad.98sf7m0hh |

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
