## [Decision Letter]

**Acceptance summary:**

The antibacterial mechanism of the cyclic peptide antibiotic colistin has been controversial. This paper is of interest to scientists within the field of antimicrobial discovery, as it implies a major adjustment to our current understanding of colistin's mechanism of action and translates this knowledge to potential therapeutic applications.

**Decision letter after peer review:**

Thank you for submitting your article "Colistin kills bacteria by targeting lipopolysaccharide in the cytoplasmic membrane" for consideration by *eLife*. Your article has been reviewed by 3 peer reviewers, and the evaluation has been overseen by myself. The following individuals involved in review of your submission have agreed to reveal their identity: Lianghui Gao (Reviewer #2); Willem van Schaik (Reviewer #3).

Summary:

The submitted manuscript presents an argument for a novel mechanism of action for the antibiotic colistin. The authors suggest that colistin kills bacteria through its action on lipopolysaccharides at the inner membrane. This is primarily supported by MCR-1 mediated colistin resistance conferring resistance only to cell lysis and not to outer membrane permeabilization. The authors extend this hypothesis to suggest that increasing the amount of LPS in the inner membrane should increase susceptibility to colistin. By inhibiting LPS transport with murepavadin, the accumulation of LPS in the cytoplasmic membrane increased. Combinations of colistin and murepavadin act synergistically to improve bacterial lysis and show efficacy in a murine lung infection model.

Essential Revisions:

1. Observations demonstrating MCR-1 modification does not impact outer membrane perturbation and provides resistance to colistin induced lysis are supported by MacNair et al. They suggest that strengthened LPS packing provided by mcr-1 could play an important role in reducing the uptake and lytic activities of colistin. The author's should address that decreased colistin uptake could also result in reduced lysis. To support their hypothesis, the relationship between the amount of modified LPS in the inner membrane and resistance to cell lysis could be expanded on. https://www.nature.com/articles/s41467-018-02875-z

2. The authors use the lack of change in susceptibility of mcr-1 spheroplasts to daptomycin and nisin to support that there is no change to the biophysical properties of the phospholipid bilayer of the cytoplasmic membrane. However, whether the sensitivity of daptomycin and nisin to changes to membrane charge or fluidity remains unclear.

3. Murepavadin is used to increase LPS at the CM and as interpreted would support the hypothesis. However, it is also possible that in the whole cell assays, the OM disruption of colistin sensitizes the cells to the killing activity of murepavadin. Repeating the assays with a non-lethal OM permeabilizer like polymyxin B nonapeptide would eliminate this possibility and strengthen the authors conclusions.

4. The authors suggest that mcr-1 provides protection from colistin through the modification of LPS at the inner membrane and that outer membrane modification has no impact on colistin activity. In contrast, it has been demonstrated that mcr-1 decoration is capable of preventing outer membrane perturbation by polymyxin B nonapeptide (https://www.nature.com/articles/nmicrobiol201728). This suggests that modified LPS at both the inner and outer membrane may play a role in resistance.

5. Authors should discuss work in the synergy between novobiocin and colistin where novobiocin enhances colistin killing through the stimulation of LPS transport. https://www.ncbi.nlm.nih.gov/pmc/articles/PMC5990483/

6. Hydrophobic NPN dye was used to explore the permeabilization of OM in this work. However, the uptake of NPN is not absolute proof that colistin is permeable. The authors should discuss this as a possible caveat of their mechanistic model.

---

## [Author Response]

Essential Revisions:1. Observations demonstrating MCR-1 modification does not impact outer membrane perturbation and provides resistance to colistin induced lysis are supported by MacNair et al. They suggest that strengthened LPS packing provided by mcr-1 could play an important role in reducing the uptake and lytic activities of colistin. The author's should address that decreased colistin uptake could also result in reduced lysis. To support their hypothesis, the relationship between the amount of modified LPS in the inner membrane and resistance to cell lysis could be expanded on. https://www.nature.com/articles/s41467-018-02875-z

We thank the reviewers for raising this important point. We agree that it is possible that whilst the OM of *E. colimcr*-1 is permeabilised by colistin, the modified LPS might restrict access of the antibiotic to the periplasm and thereby reduce killing and lysis. We have now made this point clear in the revised manuscript (please see lines: 190-194). However, since much of our data comes from studies with spheroplasts this possibility does not invalidate our findings or conclusion that colistin targets LPS in the CM and that this interaction is required for bacterial killing and lysis of susceptible strains.

To investigate the relationship between the amount of modified LPS in the inner membrane and resistance, we grew our *E. coli mcr*-1 strain with or without IPTG to modulate the levels of MCR-1 and thus the degree of LPS modification at the CM. We then generated spheroplasts of these bacteria and from *E. coli* pEmpty where none of the LPS is modified and exposed them to colistin. This revealed a clear relationship between the abundance of unmodified LPS in the CM and susceptibility to colistin-mediated permeabilization and lysis (please see Figure 2—figure supplement 2) (please see lines: 216-219).

2. The authors use the lack of change in susceptibility of mcr-1 spheroplasts to daptomycin and nisin to support that there is no change to the biophysical properties of the phospholipid bilayer of the cytoplasmic membrane. However, whether the sensitivity of daptomycin and nisin to changes to membrane charge or fluidity remains unclear.

We agree that this is an important point. To address definitively whether modification of LPS by MCR-1 affects the biophysical properties of the cytoplasmic membrane, we have assessed both membrane fluidity and membrane charge. These assays demonstrate that MCR-1-mediated LPS modification does not affect CM fluidity and has only a slight effect on CM charge, in keeping with the relatively small amount of LPS in the CM (please see Figure 2—figure supplement 1 and lines: 195-201 in the revised manuscript).

3. Murepavadin is used to increase LPS at the CM and as interpreted would support the hypothesis. However, it is also possible that in the whole cell assays, the OM disruption of colistin sensitizes the cells to the killing activity of murepavadin. Repeating the assays with a non-lethal OM permeabilizer like polymyxin B nonapeptide would eliminate this possibility and strengthen the authors conclusions.

To test whether the synergy between colistin and murepavadin is due to OM permeabilization caused by the polymyxin, we undertook two experiments:

1. As suggested by the reviewers, we have assessed whether murepavadin synergises with polymyxin B nonapeptide in checkerboard assays. By contrast to colistin and murepavadin, which show synergy, murepavadin and polymyxin B nonapeptide were not synergistic (please see Figure 4—figure supplement 5, figure supplement 6).

2. We pre-treated *P. aeruginosa* with murepavadin alone to cause LPS accumulation in the CM, then removed the murepavadin by washing before exposing the bacteria to colistin alone. The murepavadin-treated cells were much more susceptible to colisitin than untreated cells (please see Figure 4—figure supplement 7).

Taken together, the data from these two experiments confirm that the synergy observed between murepavadin and colistin is not due to OM permeabilization by the polymyxin antibiotic. Rather, it is due to the murepavadin-mediated accumulation of LPS in the cytoplasmic membrane increasing the susceptibility of bacteria to colistin (please see lines: 295-308).

4. The authors suggest that mcr-1 provides protection from colistin through the modification of LPS at the inner membrane and that outer membrane modification has no impact on colistin activity. In contrast, it has been demonstrated that mcr-1 decoration is capable of preventing outer membrane perturbation by polymyxin B nonapeptide (https://www.nature.com/articles/nmicrobiol201728). This suggests that modified LPS at both the inner and outer membrane may play a role in resistance.

We would like to make clear that we don’t claim that modification of LPS at the OM has no impact on colistin activity, just that it doesn’t prevent permeabilization of the OM, which is in keeping with previous work by Macnair et al., 2018. We have edited the text to make this point clear (please see lines: 190-191).

5. Authors should discuss work in the synergy between novobiocin and colistin where novobiocin enhances colistin killing through the stimulation of LPS transport. https://www.ncbi.nlm.nih.gov/pmc/articles/PMC5990483/

The revised manuscript now contains a discussion of this work, which shows that increased LPS transport to the outer membrane increases colistin susceptibility (please see lines: 455-466).

6. Hydrophobic NPN dye was used to explore the permeabilization of OM in this work. However, the uptake of NPN is not absolute proof that colistin is permeable. The authors should discuss this as a possible caveat of their mechanistic model.

To confirm the permeabilization of the outer membrane of MCR-1 expressing *E. coli* by colistin, we have confirmed the synergy between colistin and rifampicin in checkerboard assays as described by Macnair et al., 2018. Since rifampicin cannot normally cross the outer membrane, these data and those of Macnair et al., provide additional evidence for outer membrane disruption of *E. coli* MCR-1 cells by colistin (please see lines: 149-155 and Figure 1—figure supplement 6).